# Horizontally transferred cell-free chromatin particles function as autonomous 'satellite genomes' and vehicles for transposable elements within host cells

**Soumita Banerjee[1,2†], Soniya Sanjay Shende[1,2†], Laxmi Kata[1,2†], Relestina Simon Lopes[1,2†], Swathika Praveen[1,2], Ruchi Joshi[1,2], Naveen Kumar Khare[1,2], Gorantla V Raghuram[1,2], Snehal Shabrish[1,2], Indraneel Mittra[1,2]\***

[1]Translational Research Laboratory Advanced Centre for Treatment, Research and Education in Cancer Tata Memorial Centre Kharghar, Navi Mumbai, India; [2]Homi Bhabha National Institute, Anushakti Nagar, Mumbai, India

**\*For correspondence:** indraneel.mittra@gmail.com

[†]These authors contributed equally to this work

**Competing interest:** The authors declare that no competing interests exist.

## eLife Assessment

The authors examine the effect of cell-free chromatin particles (cfChPs) derived from human serum or from dying human cells on mouse cells in culture and propose that these cfChPs can serve as vehicles for cell-to-cell active transfer of foreign genetic elements. The work presented in this paper is intriguing and potentially **important**, but it is **incomplete**. At this stage, the claim that horizontal gene transfer can occur via cfChPs is not well supported because it is only based on evidence from one type of methodological approach (immunofluorescence and fluorescent in situ hybridization (FISH)) and is not validated by whole genome sequencing.

**Abstract** Horizontal gene transfer (HGT) plays an important evolutionary role in prokaryotes, but it is less frequent in mammals. We previously reported that cell-free chromatin particles (cfChPs) - chromosomal fragments released from the billions of dying cells that circulate in human blood - are horizontally transferred to healthy cells with biological effects. However, the underlying mechanism and function of these effects remained unclear. We treated NIH3T3 mouse fibroblasts cells with cfChPs isolated from human serum and serially passaged the cells. The intracellular activities of cfChPs were analysed using chromatin fibre fluorography, cytogenetic analysis, immunofluorescence, and fluorescent in situ hybridisation. We discovered that the internalised cfChPs were almost exclusively comprised of non-coding DNA, and the disparate DNA sequences contained within them had randomly combined to form complex concatemers, some of which were multi-mega base pairs in size. The concatemers autonomously performed many functions attributable to the nuclear genome such as DNA, RNA and protein synthesis. They harboured human LINE-1 and *Alu* elements, with the potential to rearrange themselves within the mouse genome. Our results suggest that a cell simultaneously harbours two autonomous genome forms: one that is inherited (hereditary genome) and numerous others that are acquired (satellite genomes). The satellite genomes may have evolutionary functions given their ability to serve as vehicles for transposable elements and to generate a plethora of novel proteins. Our results also suggest that 'within-self' HGT may occur in mammals on a massive scale via the medium of cfChP concatemers that have

undergone extensive and complex modifications resulting in their behaviour as 'foreign' genetic elements.

## Introduction

Horizontal gene transfer (HGT) plays an important role in the adaptation of microorganisms to changing environments and in evolutionary processes (*Bushman, 2002*). However, defining the occurrence of HGT in mammals has been a challenge. Nonetheless, experimental studies on HGT have offered insight into the regulations and functions of exogenously introduced genes and facilitated the design of drug-resistant cells, transgenic plants, and animals (*Hofmann et al., 2004*; *Nagler et al., 2011*; *Puonti-Kaerlas et al., 1990*). Indeed, gene therapy is founded on the principle of HGT, and several studies have indicated that oncogenes can be horizontally transferred across mammalian cells (*Anker et al., 1994*; *Bergsmedh et al., 2001*; *Dvořáková et al., 2013*; *García-Olmo et al., 2010*; *Trejo-Becerril et al., 2012*). These observations support the hypothesis that HGT may occur naturally in mammalian cells and contributes to evolutionary, and potentially oncogenic, processes.

Several hundred billion to trillion cells die in the human body every day (*Fliedner et al., 2002*; *Sender and Milo, 2021*) and release chromosomal fragments in the form of cell-free chromatin particles (cfChPs), which enter the extracellular compartments, including the circulation (*Holdenrieder and Stieber, 2009*). We previously reported that cfChPs circulating in the blood, and those that are released locally from dying cells, are readily internalised by living cells through horizontal transfer followed by their association with the host cell genome (*Mittra et al., 2015b*; *Mittra et al., 2017*). The latter was confirmed by whole-genome sequencing and fluorescence in situ hybridisation (FISH) analysis, as well as by the detection of multiple human *Alu* elements in the mouse recipient cells. Cellular uptake of cfChPs has several biological ramifications such as induction of DNA damage and the activation of inflammatory and apoptotic pathways (*Mittra et al., 2015b*; *Mittra et al., 2017*). Given the high turnover of cells, it can be assumed that all cells in the body continually internalise horizontally transferred cfChPs throughout their lifespan, leading to somatic mosaicism, which may be associated with ageing, chronic diseases, and cancer (*Raghuram et al., 2019*). However, the mechanisms and processes governing the intracellular consequences of horizontally transferred cfChPs remain unclear.

In this study, we investigated the intracellular processes and functions that regulate the various biological activities of cfChPs. We selected NIH3T3 mouse fibroblast cells for our experiments as we have extensive experience using this cell line for studies relating to cellular transformation, and these cells have also been widely used in various studies relating to oncogene discovery (*Parada et al., 1982*; *Santos et al., 1982*). In our previous study (*Mittra et al., 2015b*), we investigated the relative biological activities and functions of healthy and cancerous cfChPs. We found that cfChPs isolated from cancer patients had significantly greater activity in terms of DNA damage and activation of apoptotic pathways than those isolated from healthy individuals. We have also reported that cfChPs released locally from dying cells can horizontally transfer themselves into healthy bystander cells, leading to DNA damage and inflammation (*Mittra et al., 2017*). Therefore, in the present study, we treated NIH3T3 cells with cfChPs isolated from the sera of both cancer patients and healthy individuals, as well as with those released from hypoxia-induced dying MDA-MB-231 human breast cancer cells. The cells were serially passaged following cfChPs treatment, and using chromatin fibre fluorography, cytogenetic analysis, and treatment with appropriate antibodies and fluorescent in situ hybridisation (FISH) probes, we analysed the intracellular dynamics and processes that regulate the functional properties of the internalised cfChPs. The results of these experiments are presented in this article, which suggests that they may help to throw new light on mammalian evolution, ageing and cancer.

**Cell death, horizontal transfer of cell-free chromatin and mammalian evolution**

**Video 1.** A complete summary of our work.

https://elifesciences.org/articles/103771/figures#video1

## Results

The results of the study are summarized in *Video 1*.

Our results are described under three separate headings depending on whether NIH3T3 cells were treated with cfChPs isolated from sera of cancer patients or healthy individuals or with cfChPs that were released from hypoxia-induced dying MDA-MB-231 breast cancer cells. Most of the experiments were conducted using cfChPs isolated from cancer patients because of our special interest in cancer and our earlier results (*Mittra et al., 2015b*), which had shown that cfChPs isolated from cancer patients had significantly greater activity in terms of DNA damage and activation of apoptotic pathways than those isolated from healthy individuals. The majority of experiments were performed more than once with reproducible results.

### Experiments using cfChPs from cancer patients
### cfChPs are rapidly and abundantly internalised by NIH3T3 mouse fibroblast cells

We isolated cfChPs from the sera of cancer patients by a protocol described by us earlier (*Mittra et al., 2015b*). A representative electron microscopy image of the cfChPs is given in *Figure 1*. In order to track the intracellular activities of cfChPs, we fluorescently dually labelled the isolated cfChPs in their DNA with Platinum Bright Red 550 and in their histones with ATTO-488 and applied them (10 ng) to NIH3T3 mouse fibroblast cells. A representative image of isolated cfChPs that had been fluorescently dually labelled and used to treat the NIH3T3 cells is given in *Figure 2a*. Examination of NIH3T3 mouse fibroblasts treated with the dually fluorescently labelled cfChPs confirmed our earlier observation (*Mittra et al., 2015b*) that cfChPs can readily and rapidly horizontally transfer themselves to healthy cells via phagocytosis (*Mittra et al., 2017*) to accumulate in their cytoplasm and nuclei by 6 hr (*Figure 2b*). The fluorescent cfChPs appeared prominent, ostensibly owing to the amalgamation of multiple cfChPs to form large concatemers following their cellular entry (described in detail later). Chromatin fibres prepared from similarly treated NIH3T3 cells at 6 hr revealed the presence

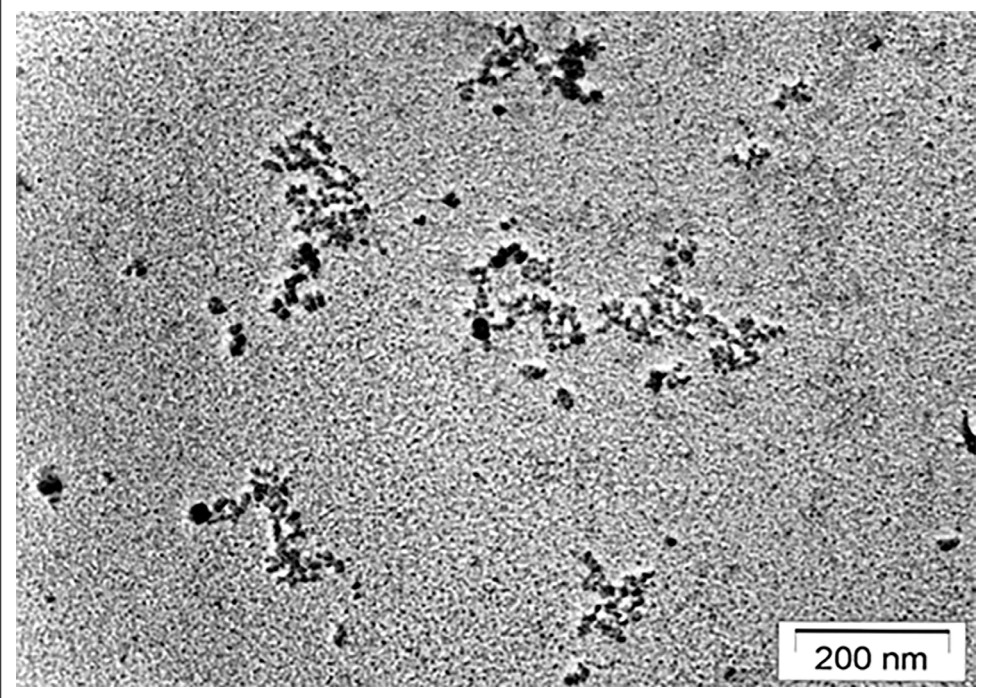

**Figure 1.** Electron microscopy image of cfChPs isolated from pooled serum of patients with cancer. A 'beads-on-a-string' appearance typical of chromatin is clearly seen. Reproduced with permission from *Mittra et al., 2015b*.

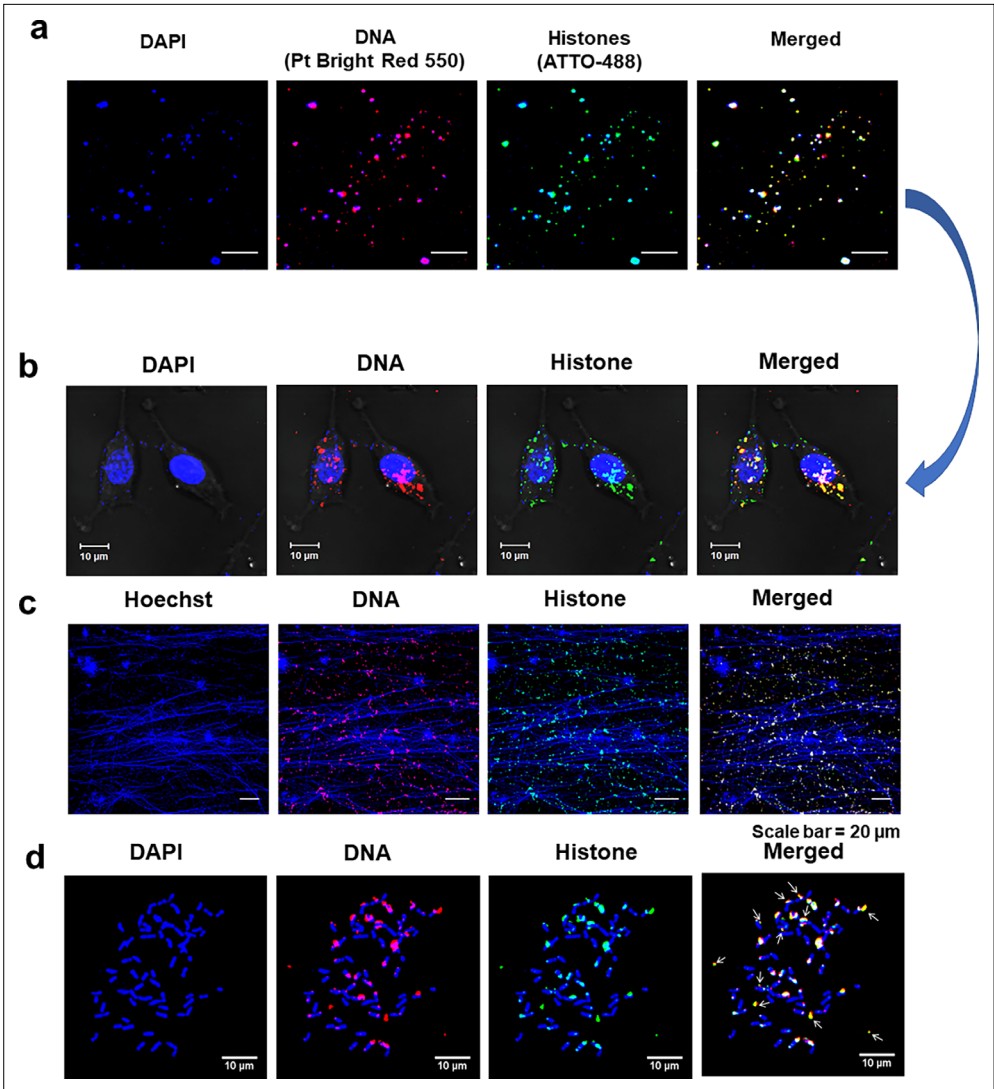

**Figure 2.** Abundant uptake of cfChPs by NIH3T3 cells at 6 hr. The DNA and histones of cfChPs were dually fluorescently labelled with Platinum Bright 550 and ATTO-488, respectively, and applied (10 ng) to NIH3T3 cells. (**a**) Representative images of dually labelled cfChPs prior to their application to NIH3T3 cells. (**b**) Confocal microscopy images of cfChPs-treated cells at 6 hr showing many dually labelled fluorescent signals in the cytoplasm and nuclei. (**c**) Fluorescence microscopy images of chromatin fibres prepared from similarly treated cells at 6 hr showing numerous dually labelled fluorescent signals of varying sizes in the cytoplasm and in association with the chromatin fibres. (**d**) Fluorescence microscopy images of metaphase spreads prepared from cfChPs-treated cells at 6 hr showing multiple dually labelled fluorescent signals, which are either associated with the chromosomes or are present in extrachromosomal spaces. The latter are marked with arrows.

of numerous dually labelled cfChPs of varying sizes, which were present in both the cytoplasm and overlapping with mouse chromatin fibres (*Figure 2c*). Since chromatin fibres are derived from inter-phase cells, we further investigated whether the internalised concatemers could also be detected in metaphase cells. The detection of concatemers on metaphase spreads was of especial interest to us in view of the resurgence of interest in extra-chromosomal DNA (ecDNA) in human cancers, which have largely been conducted on metaphase chromosomes (*Wu et al., 2022*; *Yan et al., 2024*). We were interested to examine how the cfChP concatemers might relate to the reported characteristics of cancer ecDNAs. Metaphase spreads prepared from cfChPs-treated NIH3T3 cells at 6 hr revealed many dually labelled cfChPs, some of which were associated with the chromosomes while others were present in extrachromosomal spaces (*Figure 2d*).

## cfChPs randomly combine to form complex concatemers

We had earlier hypothesised that when the cfChPs are horizontally transferred to another cell, the latter perceives the dsDNA breaks present in their two ends as damaged 'self' DNA, and in an attempt to repair the 'perceived damage', activates proteins of the DDR pathway which links up multiple disparate cfChPs as a part of the repair process leading to the formation of concatemers of variable sizes (*Mittra et al., 2015b*). To test this hypothesis, we performed FISH analysis using multiple different pairs of human chromosome-specific FISH probes on chromatin fibres and metaphase spreads prepared from cfChPs-treated NIH3T3 cells that were in continuous passage. Since these chromosome- specific probes were custom synthesised, we ensured that the probes that we used were human-specific and did not cross-react with mouse chromosomes (*Figure 3—figure supplement 1a and b*). We detected frequent co-localisation of red and green fluorescent signals suggesting that unrelated chromosomal fragments containing disparate DNA sequences had randomly amalgamated with each other to form highly complex concatemers (*Figure 3*). Some of the concatemers appeared strikingly prominent, which is particularly evident from images presented in *Figure 3a*. This suggested that the components of the concatemer had been markedly amplified. Even the small fluorescent signals shown in *Figure 3b* appeared as concatemers. As the analytical resolution of a FISH signal under a fluorescent microscope is in the range of 100–200 Kb (*Cui et al., 2016*), even the small concatemers can be assumed to be at least of a similar size range. Surprisingly, we detected the fluorescence signals of chromosome 4 to co-localise with those of centromeres and of chromosome 22 to co-localise with those of telomeres. This finding highlighted the extent of genetic complexity and chaotic nature of the concatemers (*Figure 3a*, panels 4 and 5). Perhaps the ultimate evidence of the chaotic composition of the concatemers comes from our detection of co-localising signals of telomeres and centromeres (*Figure 3a*, panel 6). The centromeric signal seen in *Figure 3a*, panel 4 is strikingly prominent, suggesting that the concatemers had not only incorporated centromeric DNA sequences (171 bp in size) within their folds but that they had undergone large-scale amplification. Results of quantitative analysis of the degree of co-localisation of fluorescent signals generated upon treatment with FISH probes specific to different chromosome pairs are given in *Figure 3—figure supplement 2*. The extent of co-localisation ranged between 32.4% and 39.8%. Concatemerisation could also be detected in metaphase spreads prepared from serially passaged cfChPs-treated NIH3T3 cells (*Figure 3c*). Negative control experiments performed on native NIH3T3 cells that had not been exposed to cfChPs treatment did not react with human-specific FISH probes against chromosome 4 and chromosome 16 which were tested (*Figure 3—figure supplement 3*).

## Concatemers exhibit variable spatial relationships with mouse chromatin fibres

We next undertook immuno-FISH analysis using an antibody against histone H4 and a human whole genomic FISH probe on chromatin fibres prepared from cfChPs-treated NIH3T3 cells in continuous passage. We detected numerous dually labelled signals representing concatemers which exhibited extensive structural and size variability, as well as remarkably variable spatial relationships with mouse chromatin fibres (*Figure 4a*). Many concatemers were detected in the cytoplasm or to have aligned with long stretches of mouse DNA, while others exhibited unusual conformations or were seen to be dangling from the mouse DNA. A similar peculiar conformation was found in clone D5 which had been developed several years earlier by treatment of NIH3T3 cells with cfChPs isolated from sera of cancer patients and which had undergone numerous passages and multiple freeze–thaw cycles (*Mittra et al., 2015b*; *Figure 4b*). This finding suggested that concatemers persist across cell generations, showing remarkable stability. The fluorescent signals of the concatemers were also strikingly prominent, suggesting that they greatly exceeded the threshold of detection of FISH signals (100–200 kb; *Cui et al., 2016*). Control experiments were performed to confirm that the whole genomic DNA FISH probe used was human specific and did not cross-react with mouse (*Figure 4—figure supplement 1*).

## Concatemers synthesise DNA polymerase and can replicate

Cells in continuous passage were pulse-labelled with BrdU followed by immuno-FISH analysis using an antibody against BrdU and a human genomic DNA probe. The analysis revealed co-localisation of the fluorescent signals of BrdU and DNA, indicating that the concatemers were actively synthesising DNA (*Figure 5a*, upper panel). Immuno-FISH experiments further revealed that the concatemers expressed

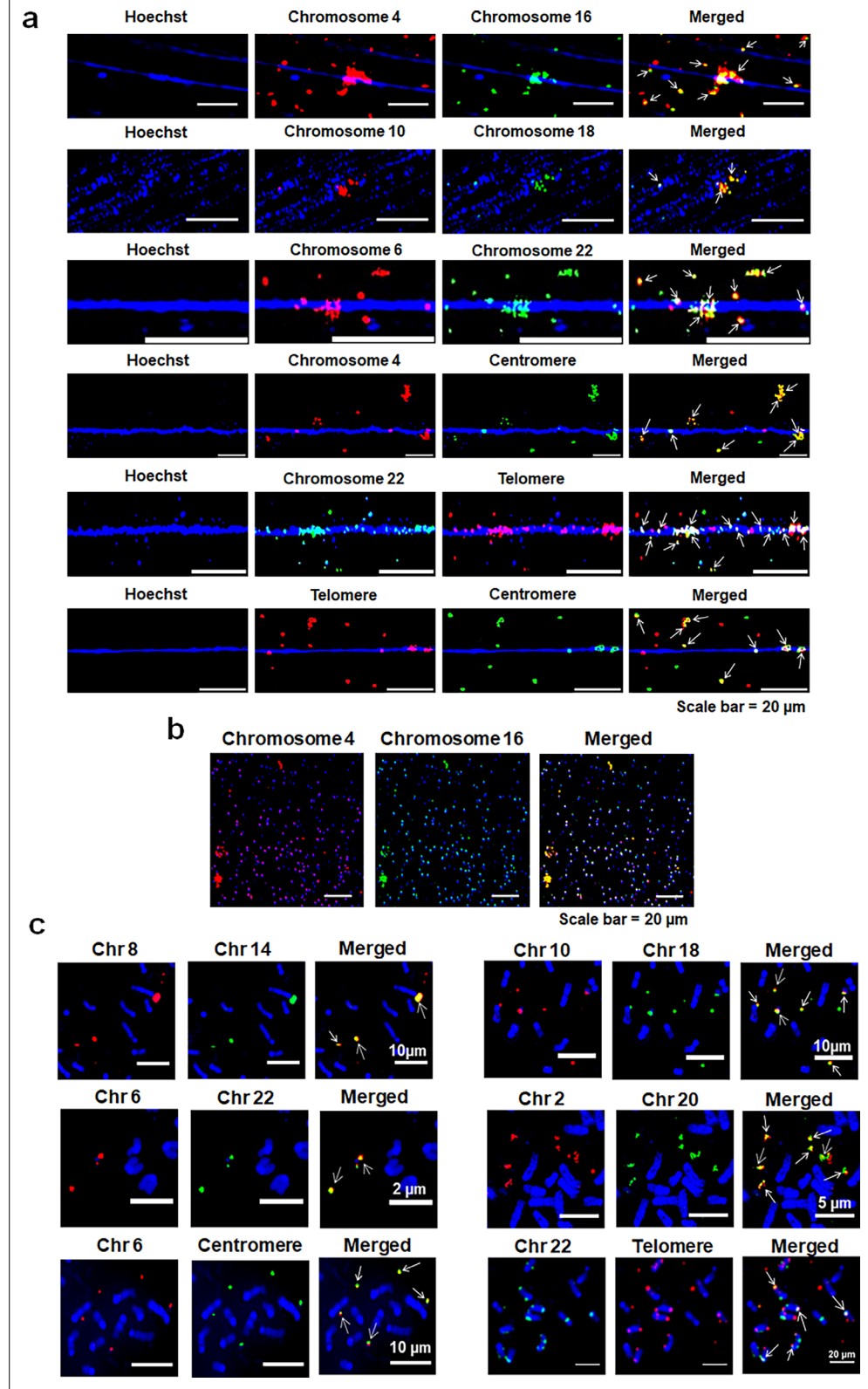

**Figure 3.** Internalised cfChPs combine to form complex concatemers. (**a**) FISH analysis of chromatin fibres prepared from cfChPs-treated NIH3T3 cells in continuous passage using different combinations of human chromosome-specific FISH probes, including probes specific for the human centromere and telomere, revealing co-localised fluorescent signals (arrows). (**b**) Small cfChPs also show co-localised signals suggesting that they too

*Figure 3 continued on next page*

*Figure 3 continued*

are comprised of concatemers (**c**) Similar co-localising signals are shown on metaphase spreads. Co-localised signals are marked with arrows.

The online version of this article includes the following figure supplement(s) for figure 3:

**Figure supplement 1.** Chromosome-specific probes that were used are human specific.

**Figure supplement 2.** Histograms representing quantitative results of the degree of co-localisation of fluorescent signals (red and green) of different pairs of chromosomes.

**Figure supplement 3.** Control experiments to detect human DNA signals in control NIH3T3 cells that had not been exposed to cfChPs.

---

human DNA polymerase γ (*Figure 5a*, middle panel), while dual immunofluorescence staining using antibodies against BrdU and human-specific DNA polymerase γ revealed co-localised signals, indicating that the concatemers had the potential to autonomously replicate themselves (*Figure 5a*, lower panel). Since human DNA polymerases are well conserved in mouse and across mammals, we ensured that the antibodies against DNA polymerase γ were human specific and did not cross-react with mouse (*Figure 5—figure supplement 1*). Results of quantitative analysis of the degree of co-localisation of fluorescent signals of human DNA and BrdU; human DNA and human DNA polymerase γ; and BrdU and human DNA polymerase γ are given in *Figure 5—figure supplement 2*. The extent of co-localisation ranged between 82.0% and 88.8%. These findings were confirmed on metaphase preparations from serially passaged cfChP-treated NIH3T3 cells, wherein we detected multiple co-localising signals of human DNA and BrdU, human DNA and human DNA polymerase γ, and BrdU and human DNA polymerase γ in the extrachromosomal spaces (*Figure 5b*). Negative control experiments performed on native NIH3T3 cells that had not been exposed to cfChPs treatment did not react

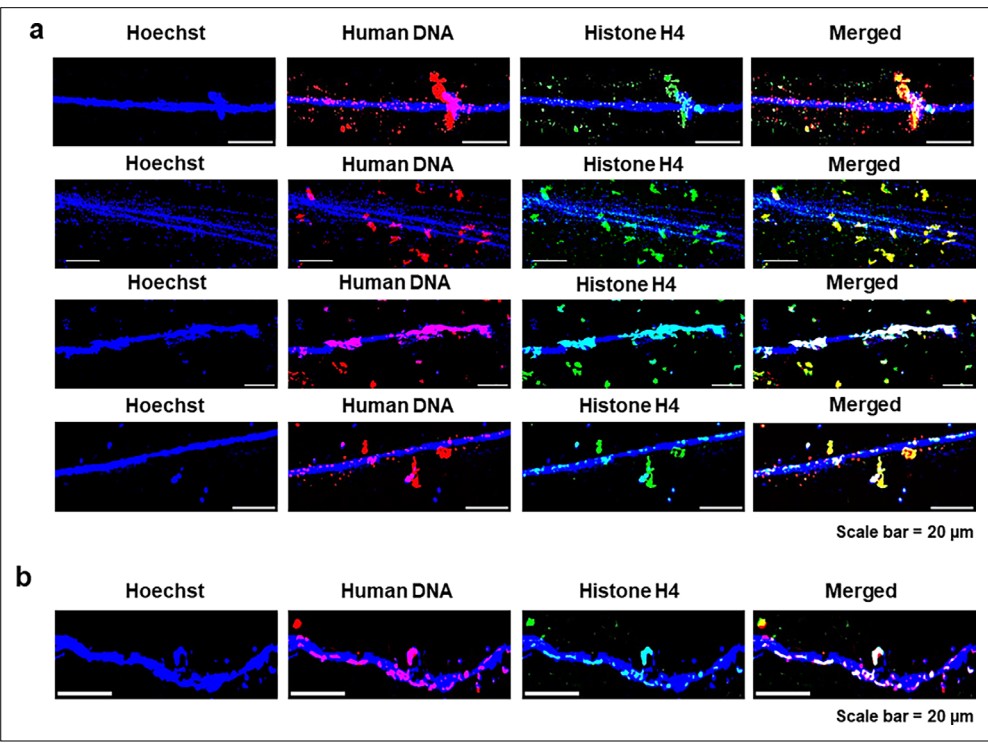

**Figure 4.** Variable spatial relationships of concatemers with mouse chromatin fibres. (**a**) Immuno-FISH analysis of chromatin fibres prepared from cfChPs-treated NIH3T3 cells in continuous passages using an antibody against histone H4 and a human whole-genome FISH probe. (**b**) Immuno-FISH analysis of chromatin fibres prepared from clone D5 showing that the concatemers persisted even after numerous passages and freeze–thaw cycles.

The online version of this article includes the following figure supplement(s) for figure 4:

**Figure supplement 1.** Control experiments to show that the genomic DNA FISH probe used was human specific and did not cross-react with mouse.

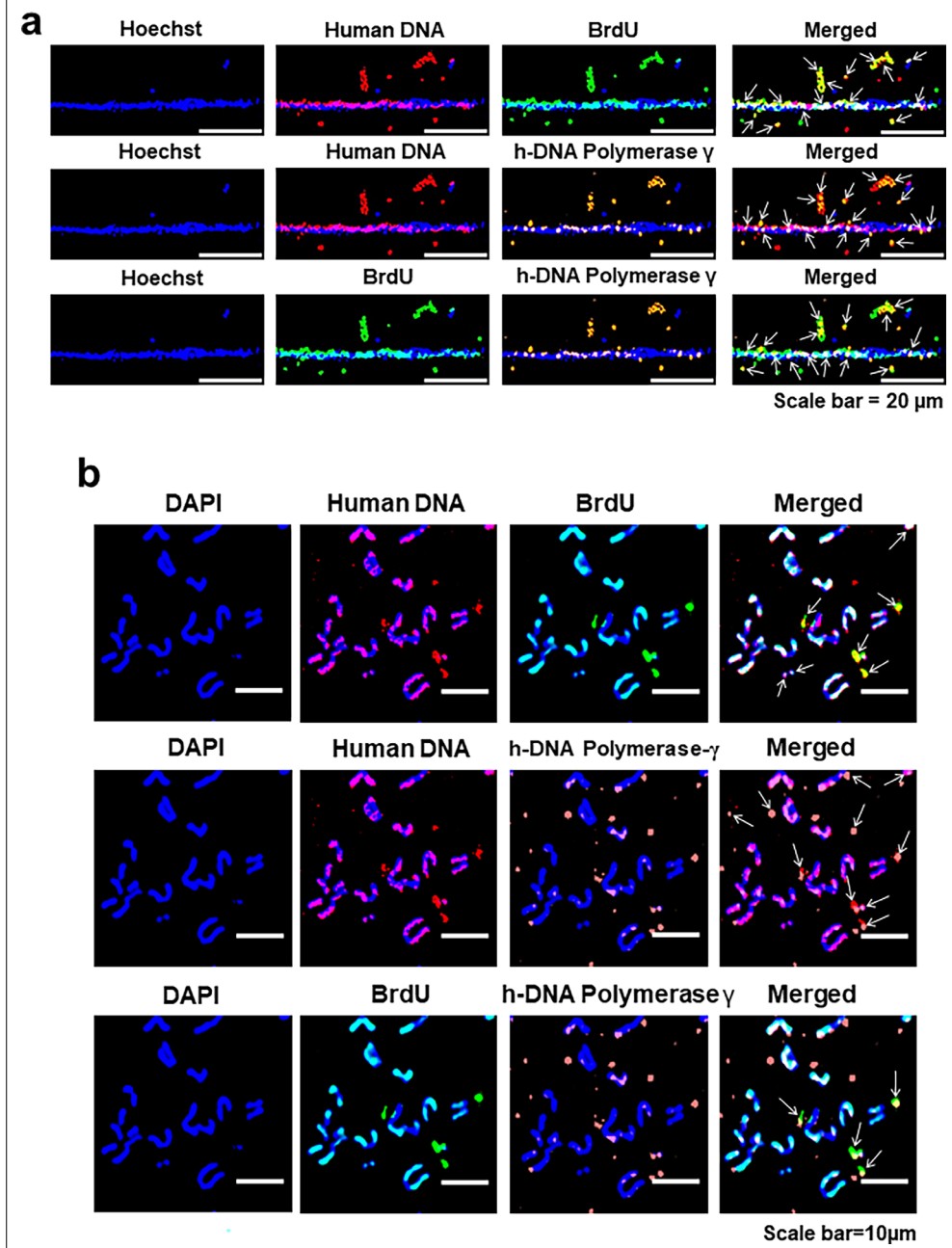

**Figure 5.** The concatemers synthesise DNA and express human DNA polymerase γ. (**a**) Chromatin fibres were prepared from NIH3T3 cells in continuous passage and were pulse-labelled for 24 hr with BrdU (10 µm). Immuno-FISH experiments using antibodies against BrdU, human-specific DNA polymerase γ, and a human whole-genome FISH probe show co-localised signals of human DNA and BrdU, human DNA and DNA polymerase γ, and BrdU and DNA polymerase γ. Co-localised signals are marked with arrows. (**b**) Similar co-localising signals are shown on metaphase spreads. Co-localised signals are marked with arrows.

The online version of this article includes the following figure supplement(s) for figure 5:

**Figure supplement 1.** Control experiments to show that the antibody used against DNA Polymerase γ was human specific and did not cross-react with mouse.

**Figure supplement 2.** Histograms representing quantitative results of the degree of co-localisation of fluorescent signals (red and green) of human DNA and BrdU, human DNA and DNA polymerase, and BrdU and DNA polymerase.

**Figure supplement 3.** Control experiments to detect human DNA signals in control NIH3T3 cells that had not been exposed to cfChPs.

with either to the human genomic DNA probe or the antibody against human DNA polymerase γ (*Figure 5—figure supplement 3*).

Given the above results that the concatemers expressed DNA polymerase which co-localised with BrdU signals, we investigated their potential for self-replication. Ten metaphase spreads were prepared from cells in each successive passage, and the number of human FISH signals (representing concatemers) per chromosome was determined. The result showed that human FISH signals on mouse chromosomes increased progressively such that the number of signals at passage 198 was 4.07 times higher than those in passage 2 (*Figure 6a*). It should be noted that this analysis is restricted to the proliferative capacity of concatemers that were associated with the chromosomes and did not take into account the replicative potential of those that are present in the cytoplasm. We also did a similar exercise to investigate whether the concatemers could amplify themselves with time by estimating the mean fluorescent intensity (MFI) per chromosome. We found that the increase in MFI between passage 2 and 198 was 237.2-fold (*Figure 6b*). Taken together, these results indicated that the concatemers increased their copy number and extensively amplified themselves with time in culture.

## Concatemers are composed of open chromatin

We next examined the epigenetic constitution of the concatemers using antibodies against the histone markers representing trimethylation of histone H3 at lysine 4 (H3K4me3) and trimethylation of histone H3 at lysine 9 (H3K9me3), indicative of regions associated with active gene promoters and regions associated with long-term repression, respectively (*Barski et al., 2007*). Chromatin fibres prepared from cfChPs-treated NIH3T3 cells in continuous passage were dually immune-stained with antibodies targeting H3K4me3 and H3K9me3, and the number of concatemers that reacted with the antibodies was counted. We found that the vast majority of the concatemers either reacted exclusively with antibodies against H3K4me3 (open chromatin) or were hybrids of H3K4me3 and H3K9me3, with only a small fraction of reacting exclusively with H3K9me3 (*Figure 7*). The figure also shows that the concatemers contained open chromatin irrespective of whether the regions of the host mouse DNA reacted with antibodies against H3K4me3 or H3K9me3. Taken together, these data indicated that concatemers primarily contained nucleosome-depleted regions and could bind to protein factors that facilitate gene transcription (*Thomas et al., 2011*) and DNA replication (*MacAlpine et al., 2010*).

## Concatemers can synthesise RNA

As the above findings suggested that concatemers are largely composed of open chromatin and potentially capable of active transcription, we investigated their potential for RNA synthesis. Using an assay kit which detects global RNA transcription, we detected abundant RNA in the cytoplasm of cfChP-treated passaged cells, which was absent in the control NIH3T3 cells (*Figure 8*). As RNA synthesis is normally restricted to the nucleus, the detection of RNA in the cytoplasm indicated that DNA contained within the concatemers was undergoing active transcription. Treatment of the cfChPs-treated passaged cells with actinomycin D or maintaining the cells at low temperature (31 °C) abolished RNA synthesis. These data indicated that the concatemers were actively involved in RNA synthesis, which is dependent on active cellular metabolism.

## Concatemers synthesise their own protein synthetic machinery

The ability of the concatemers to synthesise RNA led us to investigate whether they were involved in protein synthesis. We looked for three critical components of protein synthetic machinery, viz. ribosomal RNA, RNA polymerase, and ribosomal protein. For detection of ribosomal RNA, we under-took dual-FISH using a human genomic FISH probe (to detect the concatemers) and a FISH probe against human ribosomal RNA. We ensured that the latter FISH probe was specific to human and did not cross-react with mouse (*Figure 9—figure supplement 1*, upper panel). Dual FISH analysis on chromatin fibres prepared from serially passaged cfChP-treated cells revealed strictly co-local-ised signals, indicating that the concatemers were synthesising ribosomal RNA (*Figure 9a*, upper panel). We next investigated whether the concatemers could synthesise the other two components of the protein synthetic machinery, viz. RNA polymerase and ribosomal protein after ensuring that the antibodies against them were specific to humans (*Figure 9—figure supplement 1*, middle and lower panels). Immuno-FISH analysis revealed co-localised fluorescent signals generated by a human-specific genomic DNA probe and antibodies against RNA polymerase III and ribosomal protein

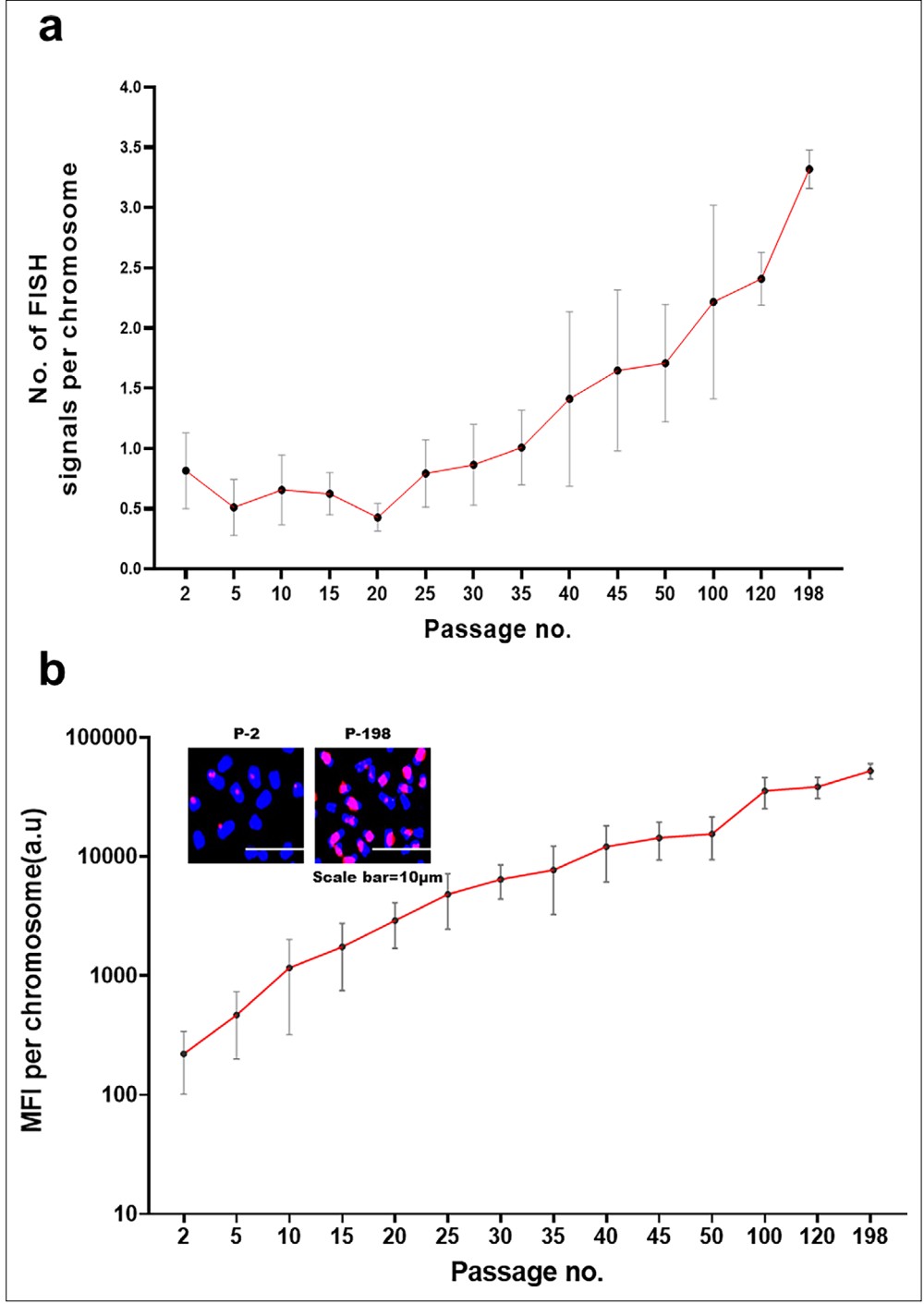

**Figure 6.** Concatemers proliferate and amplify themselves within the mouse genome over time. Metaphase spreads were prepared from cells in each progressively increasing passage and probed with a human-specific DNA FISH probe. (**a**) The total number of human DNA FISH signals on the DAPI-stained chromosomes was counted, and the mean number of human FISH signals per chromosome was calculated after analysing 10 metaphase spreads at each passage. A progressive increase in mean FISH signals per chromosomes is evident with increasing passage number (analysis of variance for linear trend p<0.0001). The mean number of human FISH signals per chromosome between passage no. 2 and passage no. 198 increased by a factor of 4.07. (**b**) A similar exercise was done as above except that mean fluorescent intensity (MFI) of human FISH signals per chromosome, indicative of amplification, was determined. A progressive increase in mean MFI per chromosomes is evident with increasing passage number (analysis of variance for linear trend p<0.0001). The MFI per chromosome increased by a factor of 237.19 between passage no. 2 and passage no. 198. The insets represent partial metaphase images showing human DNA signals on the chromosomes. Blue and red signals represent DAPI and human DNA, respectively.

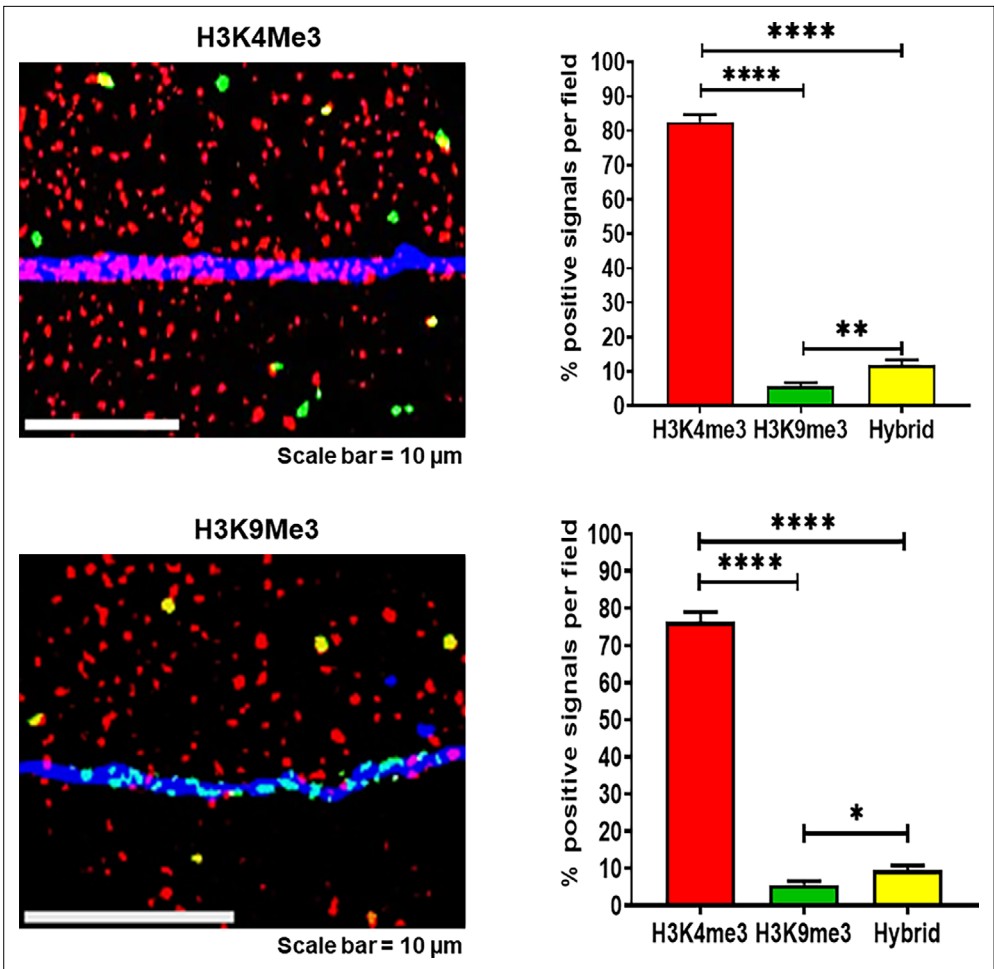

**Figure 7.** Concatemers largely comprise open chromatin irrespective of the epigenetic constitution of the host mouse DNA. Representative images of chromatin fibres immunostained with antibodies against H3k4me3 (red) and H3k9me3 (green). The hybrid concatemers are represented by co-localised yellow signals. The host mouse DNA in the upper image is seen to react with H3k4me3 antibody representing open chromatin, while the host mouse DNA segment in the lower image is comprised of heterochromatin and seen to react with antibody against H3k9me3. However, the concatemers were largely composed of open chromatin irrespective of the epigenetic status of the host mouse DNA. Histograms represent quantitative estimates of H3k4me3, H3k9me3, and hybrid histones after counting 100 fluorescent signals. The values are expressed as mean ± SEM values. Statistical analysis was performed using two-tailed Student's *t*-test. * p<0.05, ** p<0.01, and **** p<0.0001.

(*Figure 9a*, middle and lower panels, respectively). These findings indicated that concatemers could autonomously synthesise critical components of the protein synthetic machinery. Results of quantitative analysis of the degree of co-localisation of fluorescent signals of human DNA and human ribosomal RNA; human DNA and RNA polymerase III; and human DNA and ribosomal protein are given in *Figure 9—figure supplement 2*. The extent of co-localisation ranged between 77.2% and 81.4%. We confirmed these findings in metaphase preparations from serially passaged cells for all three components of the protein synthetic machinery mentioned above (*Figure 9b*). Negative control experiments performed on native NIH3T3 cells that had not been exposed to cfChPs treatment did not react with the human-specific probes against DNA, ribosomal RNA, and the human-specific antibodies against RNA polymerase III and ribosomal protein (*Figure 9—figure supplement 3*).

## Concatemers synthesise a variety of human proteins in mouse cells

Having confirmed that concatemers are capable of synthesising RNA and assembling their own protein synthetic machinery, we went on to investigate whether they were capable of autonomously

synthesising proteins. We conducted immune-FISH experiments using a human-specific genomic DNA probe (to detect the concatemers) and antibodies against various proteins. We found that the fluorescent signals of the various proteins frequently co-localised with those of human DNA, suggesting that the concatemers were capable of synthesising proteins (*Figure 10a*). Significantly, the newly synthesised proteins consistently remained associated with the concatemers of their origin (identified by fluorescent human DNA signals). This finding suggested that, although the concatemers contained the critical components of a protein synthetic machinery, they apparently lacked the machinery required for protein sorting. It should be noted that all the proteins that we detected seemed to be over-expressed, confirming that the gene segments corresponding to the proteins within the concatemers were amplified. Results of quantitative analysis of the degree of co-localisation of fluorescent signals of human DNA and of various human proteins are given in *Figure 10—figure supplement 1*. The extent of co-localisation varied between 70.0% and 72.6%. We further confirmed these findings in metaphase preparations from serially passaged cfChPs-treated NIH3T3 cells (*Figure 10b*). We also conducted extensive control experiments on native NIH3T3 cells that had not been exposed to cfChPs treatment using a variety of human-specific antibodies and found that none of them showed any positive fluorescent signals (*Figure 10—figure supplement 2*). The above results indicated that the disparate DNA sequences that comprise the concatemers are transcribed and translated to generate proteins corresponding to the diverse DNA sequences that they contain apparently using their own protein synthetic machinery. Results of control experiments confirming the human specificity of the antibodies against all the proteins described above are given in *Figure 10—figure supplement 3*.

## The proteins that concatemers synthesise are complex fusion proteins

As the concatemers are formed as a result of amalgamation of widely disparate DNA sequences, we investigated whether the proteins that they synthesised might be fusion proteins. Dual-immunofluorescence experiments using pairs of antibodies against proteins, the corresponding genes of which were located on different chromosomes, revealed that the fluorescent signals frequently

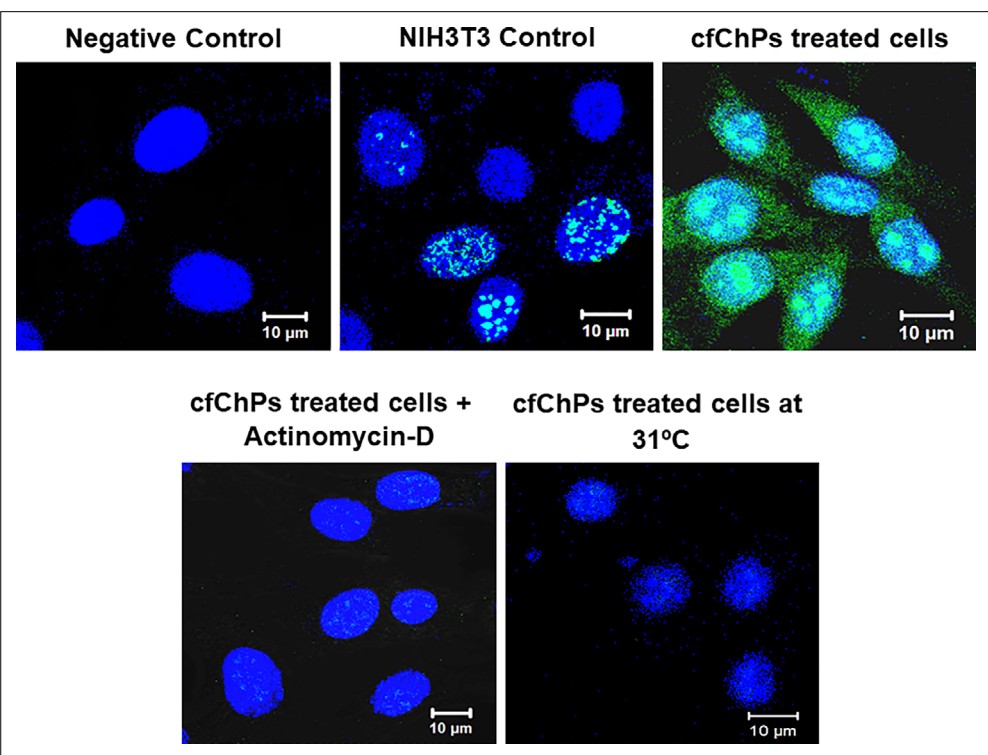

**Figure 8.** Concatemers synthesise RNA which is dependent on active cellular metabolism. Images showing cytoplasmic RNA synthesis by cfChP-treated cells in continuous passage, which is absent in control NIH3T3 cells. Treatment of cfChP-treated cells with Actinomycin D (0.0005 µg/mL) or maintenance at low temperature (31 °C) abolishes RNA synthesis indicating that the latter is dependent on active cellular metabolism.

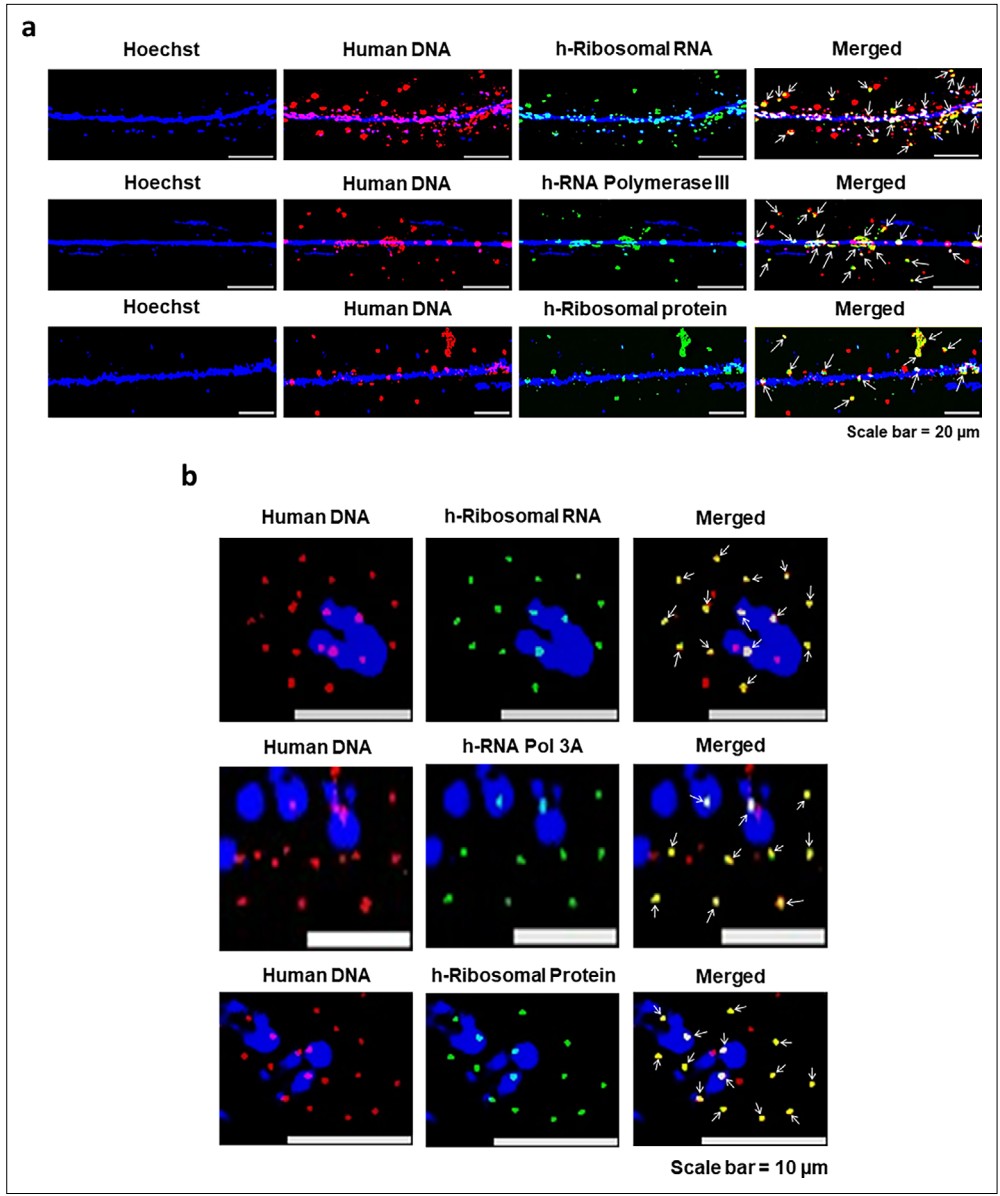

**Figure 9.** Concatemers synthesise their own protein synthetic machinery. Chromatin fibres (**a**) and metaphase spreads (**b**) were prepared from cfChPs-treated cells in continuous passage. Dual-FISH experiments were performed using a human-specific genomic DNA FISH probe and a probe against human-specific ribosomal RNA, and immune-FISH experiments were performed using a human-specific genomic DNA FISH probe and antibodies against human-specific RNA polymerase III and human-specific ribosomal protein. Co-localised fluorescent signals of DNA and the components of the above protein synthetic machinery are marked with arrows.

The online version of this article includes the following figure supplement(s) for figure 9:

**Figure supplement 1.** Control experiments to show that the FISH probes and antibodies used against the components of the protein synthetic machinery viz. ribosomal RNA, RNA polymerase, and ribosomal protein were human specific and did not cross-react with mouse.

**Figure supplement 2.** Histograms representing quantitative results of the degree of co-localisation of fluorescent signals (red and green) of human DNA and human ribosomal RNA, human DNA and human RNA polymerase III, and human DNA and human ribosomal protein.

**Figure supplement 3.** Control experiments to detect human DNA signals in control NIH3T3 cells that had not been exposed to cfChPs.

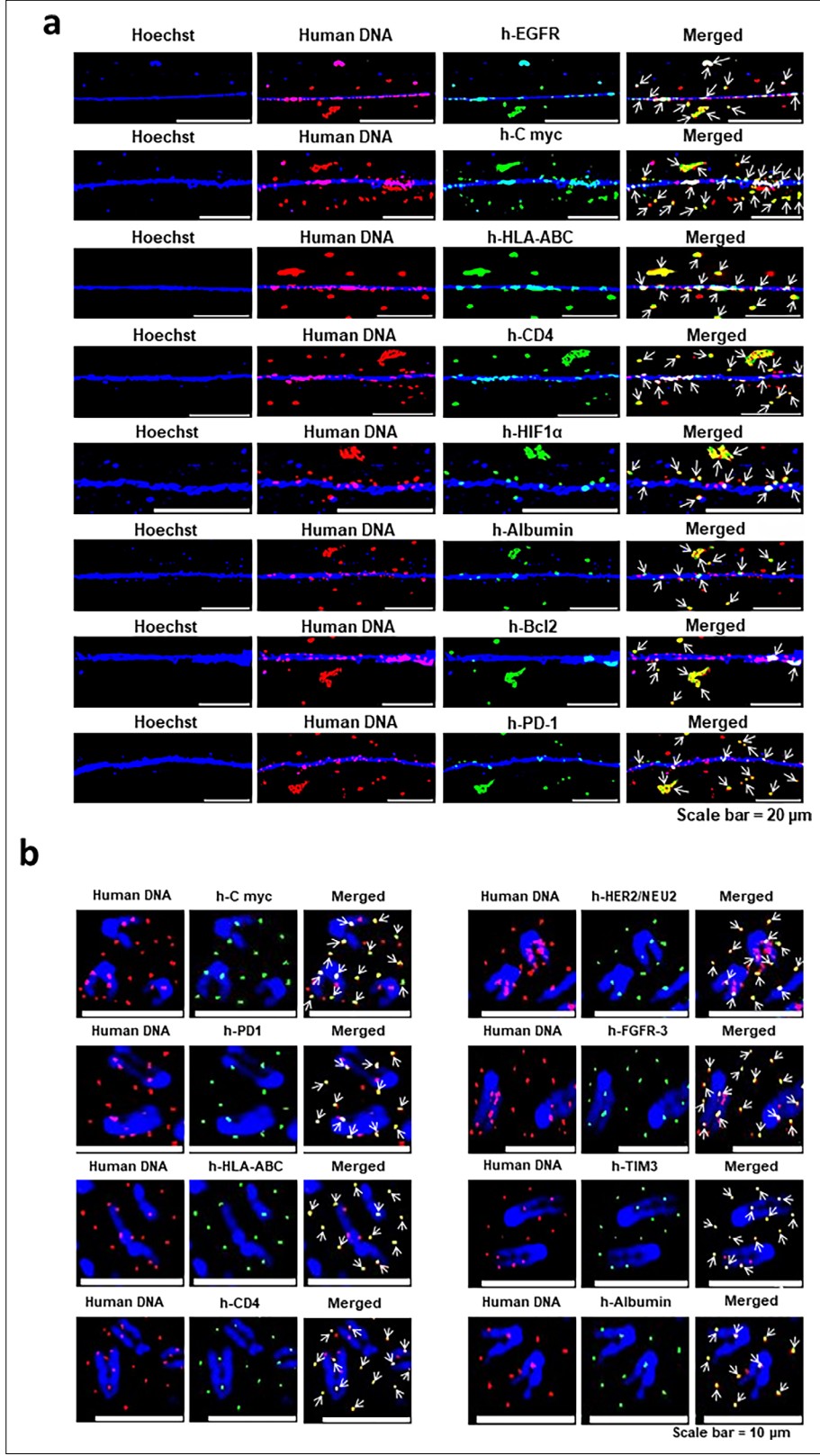

**Figure 10.** Concatemers synthesise a variety of human proteins. Immuno-FISH analysis on chromatin fibres (**a**) and metaphase spreads (**b**) prepared from cfChPs-treated NIH3T3 cells in continuous passage using human specific whole genomic FISH probe and antibodies against various human proteins. The co-localising signals of human DNA and various human proteins are marked with arrows.

*Figure 10 continued on next page*

*Figure 10 continued*

The online version of this article includes the following figure supplement(s) for figure 10:

**Figure supplement 1.** Histograms representing quantitative results of the degree of co-localisation of fluorescent signals (red and green) of human DNA and various proteins.

**Figure supplement 2.** Control experiments to detect human DNA and protein signals in control NIH3T3 cells that had not been exposed to cfChPs.

**Figure supplement 3.** Control experiments to show that the antibodies used against the various proteins were human specific and did not cross-react with mouse.

co-localised. This finding indicated that the proteins synthesised by the concatemers were fusion proteins with potentially novel functions (*Figure 11*). Fusion proteins were detected both on chromatin fibres and metaphase preparations. Results of quantitative analysis of the degree of co-localisation of fluorescent signals indicative of fusion proteins are shown in *Figure 11a*. The extent of co-localising signals ranged between 33.0% and 36.3% (*Figure 11—figure supplement 1*). Results of control experiments confirming the human specificity of the antibodies against the fusion proteins not included in *Figure 10—figure supplement 3* are given in *Figure 11—figure supplement 2*.

## Concatemers are vehicles for transposable elements

Among the fusion proteins shown in *Figure 11* we accidentally found reverse transcriptase and transposase to co-localise with Bcl2 and cMyc, respectively. This finding raised the possibility that the concatemers might harbour gene components related to transposable elements. We used human-specific LINE-1 and *Alu* DNA probes which had been custom synthesised (*Supplementary file 1*). Nonetheless, we checked for their specificity to ensure that they reacted only to human and not to mouse transposable elements (*Figure 12—figure supplement 1*). Dual-FISH experiments using a human genomic DNA probe and those targeting human LINE-1 and *Alu* showed that many of the fluorescent signals had co-localised indicating that the concatemers harboured DNA sequences of retro-transposable elements (*Figure 12a*). Results of quantitative analysis of the degree of co-localisation of fluorescent FISH signals of human DNA and human LINE-1 and Alu on chromatin fibres are given in *Figure 12—figure supplement 2*. The extent of co-localisation was 70.4% and 72.8% for LINE-1 and Alu, respectively. Similar co-localising signals were also seen on metaphase preparations (*Figure 12b*). The fact that both LINE-1 and *Alu* signals could be clearly detected by FISH analysis indicated that they were extensively amplified, given that LINE-1 and Alu are approximately 6000 bp and 300 bp in size, respectively. The issue of amplification of the transposable elements is discussed in detail under section 1.12. Control experiments performed on native NIH3T3 cells that had not been exposed to cfChPs treatment using human LINE-1 and Alu probes did not reveal any positive signals (*Figure 12—figure supplement 3*).

## LINE-1 and *Alu* elements are associated with reverse transcriptase, transposase, and DNA polymerase

Immuno-FISH analysis using antibodies against human reverse transcriptase and transposase and FISH probes against LINE-1 and *Alu* revealed co-localised signals, indicating that the enzymes reverse transcriptase and transposase were frequently associated with DNA sequences corresponding to LINE-1 and Alu (*Figure 13a and b*). Such an association could potentially allow the transposable elements to re-arrange themselves on the mouse cell genome by translocating themselves from one location to another. Results of quantitative analysis of the degree of co-localisation on chromatin fibres of fluorescent FISH signals of human LINE-1 and human reverse transcriptase and transposase; and Alu and human reverse transcriptase and transposase are given in *Figure 13—figure supplement 1*. The extent of co-localisation ranged between 34.4% and 36.1%. Control experiments performed on native NIH3T3 cells that had not been exposed to cfChPs treatment using human-specific antibodies against reverse transcriptase and transposase and FISH probes against h-LINE-1 and h-ALU did not reveal any positive signals (*Figure 13—figure supplement 2*).

As transposable elements are known to increase their copy number with time (*Li et al., 2013*), we were curious to find out whether the LINE-1 and *Alu* elements would proliferate and increase their copy number with progressively increasing passages. Immuno-FISH experiments using antibodies

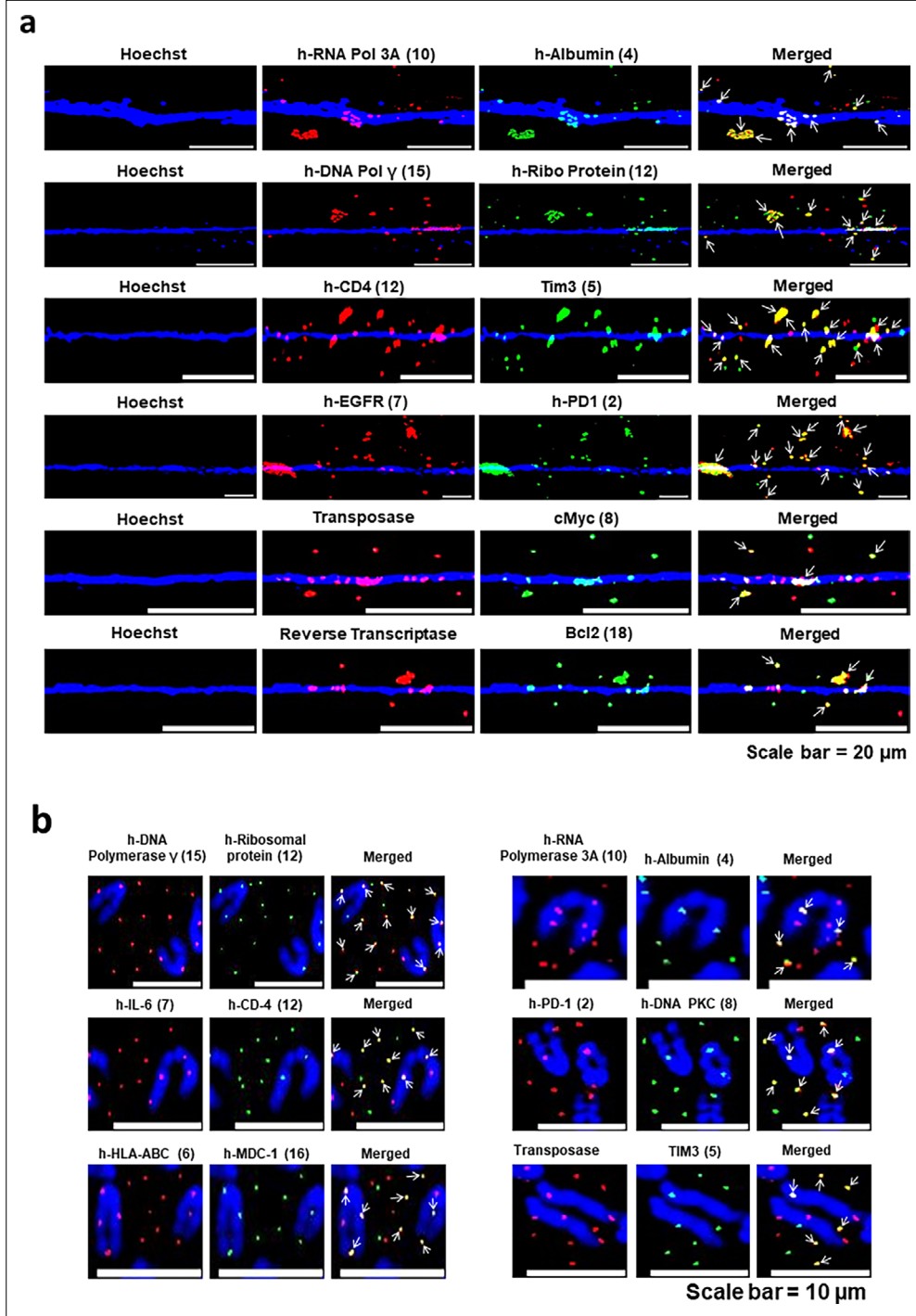

**Figure 11.** Proteins synthesised by concatemers are fusion proteins. Dual immunofluorescence analysis using antibody pairs targeting diverse human-specific proteins in chromatin fibres (**a**) and metaphase spreads (**b**) prepared from cfChPs-treated NIH3T3 cells in continuous passage. Results show frequent co-localised signals indicative of fusion proteins (arrows). The numbers given in the parenthesis indicate the chromosomal location of the genes that correspond to the proteins.

The online version of this article includes the following figure supplement(s) for figure 11:

**Figure supplement 1.** Histograms representing quantitative results of the degree of co-localisation of fluorescent signals (red and green) of different pairs of proteins.

*Figure 11 continued on next page*

*Figure 11 continued*

**Figure supplement 2.** Control experiments to show that the antibodies used against the components of fusion proteins that were not included in *Figure 10—figure supplement 3* were human specific and did not cross-react with mouse.

against human DNA polymerase γ and FISH probes against LINE-1 and *Alu* detected co-localised signals, indicating that LINE-1 and *Alu* elements were associated with DNA polymerase, raising the possibility that they may have the potential to proliferate (*Figure 14a and b*). This possibility was supported by the finding that LINE-1 and *Alu* elements could synthesise DNA. Immuno-FISH analysis of cells in continuous passage that had been pulse-labelled with BrdU showed co-localised signals generated by antibodies against BrdU and FISH probes against LINE-1 and *Alu*. Taken together, these findings supported the idea that the DNA of both the transposable elements had the potential to replicate. Results of quantitative analysis of the degree of co-localisation of fluorescent FISH signals

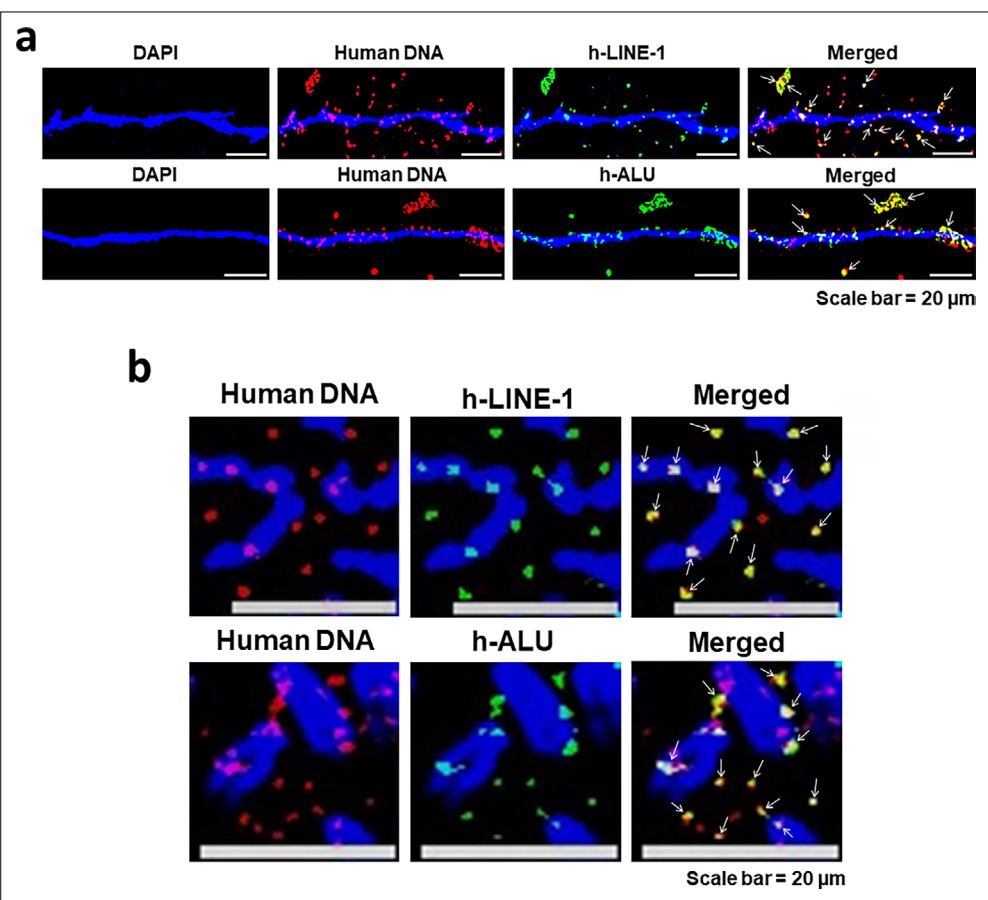

**Figure 12.** Concatemers harbour transposable elements. Dual-FISH analysis on chromatin fibres (**a**) and metaphase spreads (**b**) prepared from cfChPs-treated NIH3T3 cells in continuous passage using a human-specific genomic DNA FISH probe and those against human LINE-1 or human *Alu* show co-localised signals (marked with arrows).

The online version of this article includes the following figure supplement(s) for figure 12:

**Figure supplement 1.** Control experiments to show that the probes used against LINE-1 and *Alu* transposable elements and the antibodies against reverse transcriptase and transposase (please refer to *Figure 13* below) were human specific and did not cross-react with mouse.

**Figure supplement 2.** Histograms representing quantitative results of the degree of co-localisation of fluorescent signals (red and green) of human DNA and LINE-1, human DNA and Alu.

**Figure supplement 3.** Control experiments to detect human DNA signals in control NIH3T3 cells that had not been exposed to cfChPs.

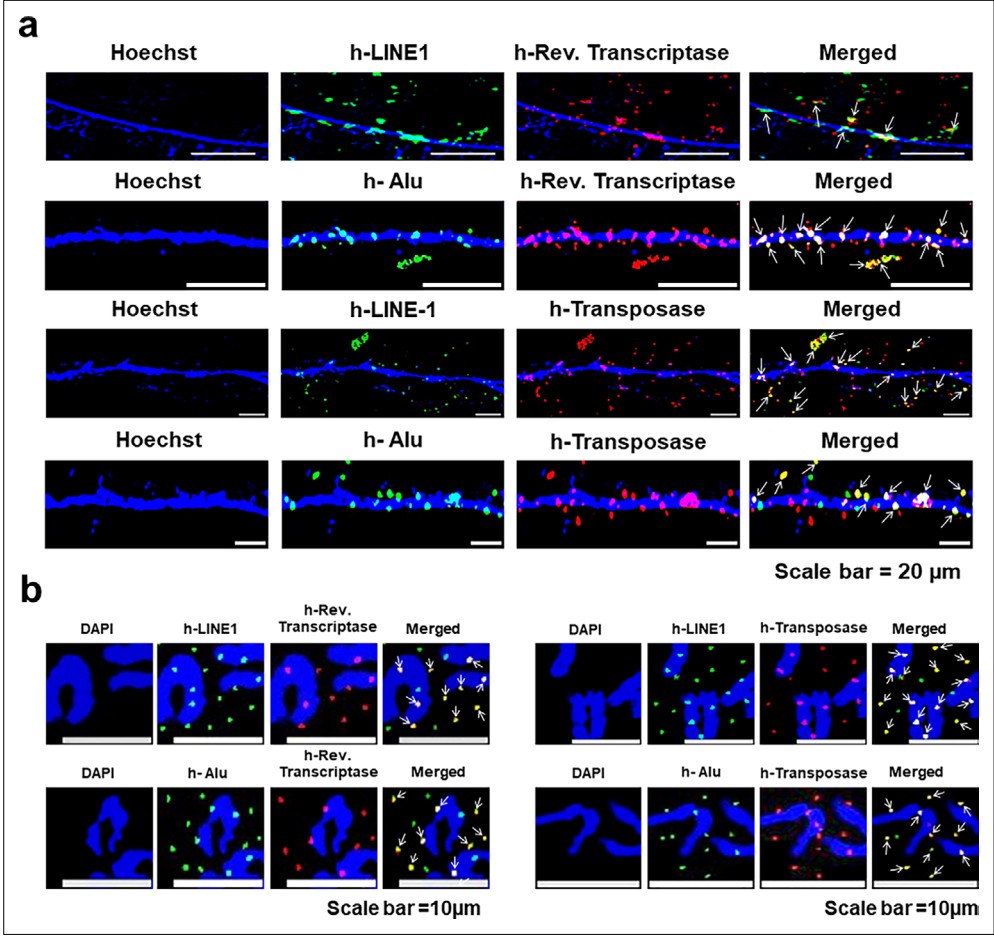

**Figure 13.** LINE-1 and Alu sequences are associated with reverse transcriptase and transposase. Immuno-FISH analysis of chromatin fibres (**a**) and metaphase spreads (**b**) prepared from cfChPs-treated NIH3T3 cells in continuous passage using human *Alu* or LINE-1 probes and antibodies against human reverse transcriptase and human transposase shows co-localising signals (marked with arrows).

The online version of this article includes the following figure supplement(s) for figure 13:

**Figure supplement 1.** Histograms representing quantitative results of the degree of co-localisation of fluorescent signals (red and green) of human LINE-1 and human reverse transcriptase, human Alu and human reverse transcriptase, human LINE-1 and human transposase, and human Alu and human transposase.

**Figure supplement 2.** Control experiments to detect human DNA and protein signals in control NIH3T3 cells that had not been exposed to cfChPs.

of human LINE-1 and human DNA polymerase and BrdU; and Alu and human DNA polymerase and BrdU are given in *Figure 14—figure supplement 1*. The extent of co-localisation varied between 35.8% and 36.9%.

## LINE-1 and *Alu* elements increase their copy number and amplify themselves with time

Given the above findings, we were curious to find out whether LINE-1 and *Alu* elements could indeed replicate and increase their copy numbers within the mouse genome. Metaphase spreads were prepared from cfChPs-treated NIH3T3 cells in continuous passage and probed with FISH probes against LINE-1 and *Alu* elements. Fifteen metaphases were analysed at each passage, and the number of LINE-1 and *Alu* fluorescent signals per metaphase was determined (*Figure 15*). Our results showed that the numbers of human LINE-1 and *Alu* signals per chromosome increased steadily between passage 2 and 198, resulting in a 7.6-fold increase in copy number in the case of LINE-1 and a 6.7-fold increase in the case of *Alu* (*Figure 15a*). These data indicated that LINE-1 and *Alu* elements progressively increased

their copy numbers by retrotransposition over time in the mouse genome. It should be noted that this analysis is restricted to the proliferative capacity of LINE-1 and *Alu* elements that are associated with chromosomes and does not take into account the replicative potential of those elements that are present in the cytoplasm. We also did a similar exercise to investigate whether the concatemers could amplify themselves by estimating the mean fluorescent intensity (MFI) of LINE-1 and *Alu* signals per chromosome. We found that the increase in MFI between passage 2 and 198 was 151.3-fold in the case of LINE-1 and 83.4-fold in the case of *Alu* (*Figure 15b*). Taken together, these data indicated that the transposable elements could not only proliferate but also extensively amplify themselves, thereby becoming increasingly effective in modifying the host genome.

## NIH3T3 cells are not unique in their ability to internalise cfChPs

We next investigated whether the ability to internalise cfChPs is a property that is unique to NIH3T3 cells or whether other cells from different species also have similar ability to internalise cfChPs. We treated four different cell lines, namely Vero (monkey kidney cells); Dolly (dog cells); B/CMBA. OV (mouse ovary cells) and HEK293 (human embryonic kidney cells) with cfChPs isolated from human serum (10 ng). The treated cells were harvested after the 5th passage, and chromatin fibres prepared from them were probed with a human whole genomic probe and a probe against human LINE-1. We detected the presence of human DNA and human LINE-1 signals in all the above cell lines indicating

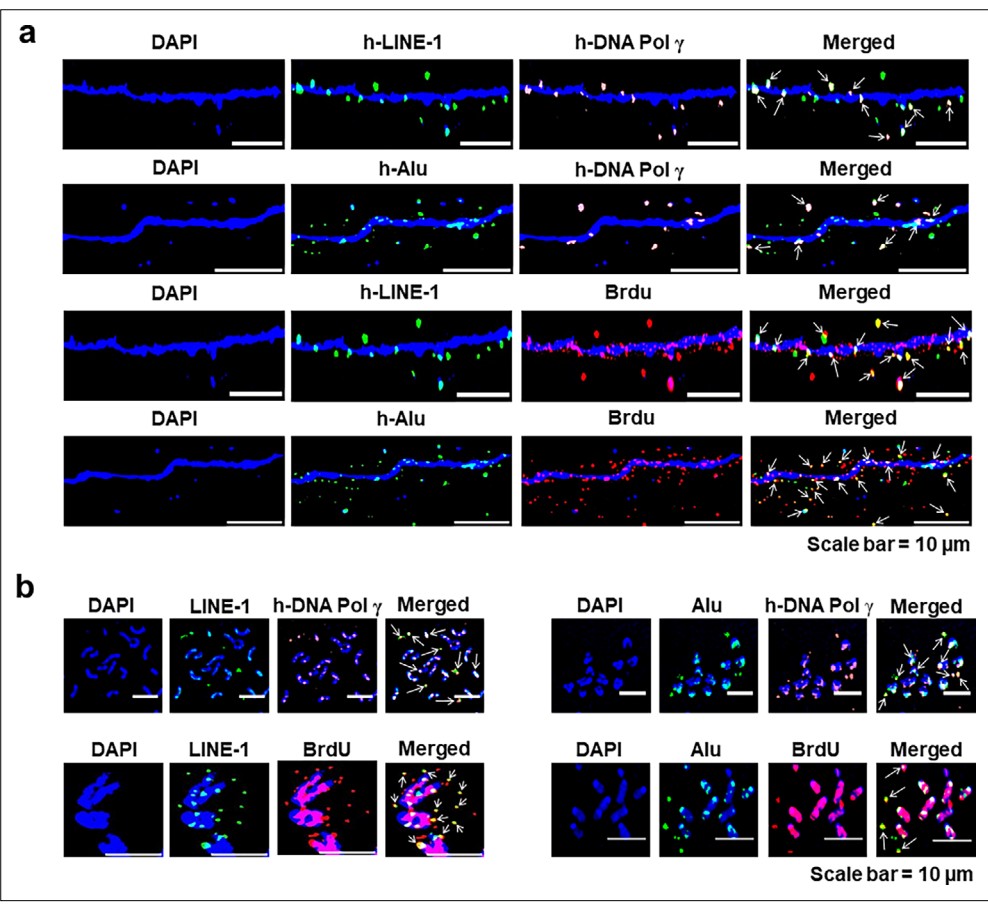

**Figure 14.** LINE-1 and *Alu* elements are associated with DNA polymerase and can actively synthesise DNA. (**a**) Immuno-FISH images of cfChPs-treated cells in continuous passage showing co-localising signals of LINE-1 (green) and Alu (green) and DNA polymerase γ (orange) and BrdU (red) on chromatin fibres. (**b**) Similar co-localising signals are shown on metaphase spreads. Co-localised signals are marked with arrows.

The online version of this article includes the following figure supplement(s) for figure 14:

**Figure supplement 1.** Histograms representing quantitative results of the degree of co-localisation of fluorescent signals (red and green) of human LINE-1 and human DNA polymerase, human Alu and human DNA polymerase, human LINE-1 and BrdU, and human Alu and BrdU.

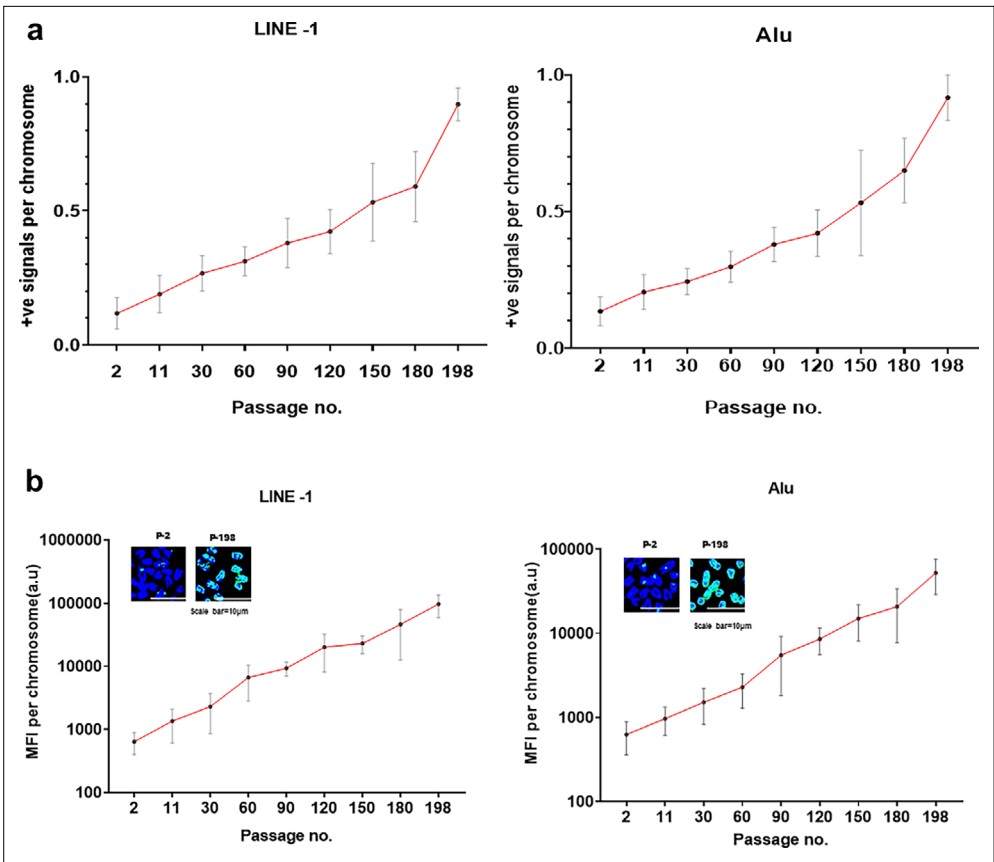

**Figure 15.** LINE-1 and *Alu* elements proliferate and amplify themselves within the mouse genome over time. Metaphase spreads were prepared from cells in each progressively increasing passage and probed with a human-specific LINE-1 and *Alu* probes. (**a**) The total number of human LINE-1 and *Alu* signals on the DAPI stained chromosomes was counted, and the mean number of LINE-1 and *Alu* signals per chromosome was calculated after analysing 15 metaphase spreads at each passage. A progressively increasing LINE-1 and *Alu* signals per chromosomes is evident with increasing passage number (analysis of variance for linear trend p<0.0001). The number of human LINE-1 signals per chromosome increased by a factor of 7.6 between passage no. 2 and passage no. 198, and by a factor of 6.7 in the case of *Alu*. (**b**) A similar exercise was done as above except that mean fluorescent intensity (MFI) of human LINE-1 and *Alu* per chromosome, indicative of amplification, was determined. A progressively increasing MFI per chromosomes is evident with increasing passage number (analysis of variance for linear trend p<0.0001). Mean MFI per chromosome increased by a factor of 151.3 between passage no. 2 and passage no. 198 in the case of LINE-1, and by a factor of 83.4 in the case of *Alu*. The insets represent partial metaphase images showing human LINE-1 and Alu signals on the chromosomes. Blue and red signals represent DAPI and human DNA, respectively.

that all of them had the ability to internalise cfChPs (*Figure 16*). This finding led us to the conclusion that horizontal transfer of cfChPs is likely to be a universal phenomenon.

## The concatemers are largely composed of non-coding DNA

Given that 99% of the human genome is comprised of non-coding DNA, and in view of our detection of LINE-1 and Alu elements among the cfChP concatemers, we investigated whether the majority of the cfChPs that had been internalised might have been derived from non-coding DNA. To investigate this, we used a human long non-coding RNA probe which comprised a set of Stellaris RNA FISH probes containing a pool of up to 48 unique sequences, each labelled with a fluorophore, that collectively bind along an RNA target transcript (*Supplementary file 1*). When we applied this RNA probe to chromatin fibres prepared from two human cell lines (HEK293 and MRC5), we found that the RNA probe aligned with the entire length of the human DNA with almost 100% coverage (*Figure 17a*). The human specificity of the RNA probe was confirmed by the demonstration that it did not bind

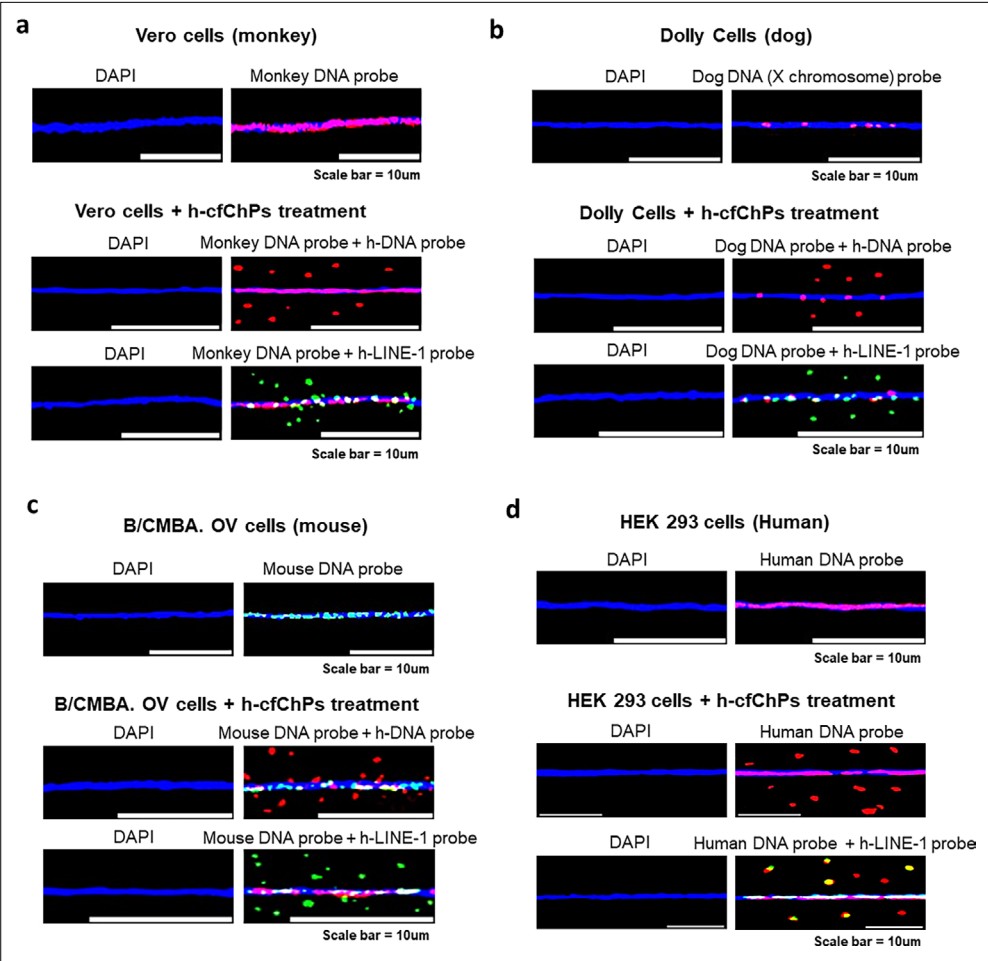

**Figure 16.** NIH3T3 cells are not unique in their ability to internalise cfChPs. We used four different cell lines other than NIH3T3 viz. Vero (monkey kidney cells); Dolly (female dog cells); B/CMBA. OV (mouse ovary cells) and HEK293 (human embryonic kidney cells) and tested them for their ability to internalise cfChPs. The cells were treated with cfChPs derived from human serum and chromatin fibres were prepared at 5th passage. The chromatin fibres were probed with a whole genomic DNA probe or with a human specific LINE-1 probe. Human DNA and LINE-1 signal are clearly seen in the treated cells. It is to be noted that the dog FISH probe was specific for X (red) and Y (green) chromosome. Since the cells came from a female dog, only the red X chromosome is seen to react.

to NIH3T3 mouse DNA (*Figure 17a*). This finding suggested that the long non-coding RNA probe could be used as a substitute to detect human non-coding DNA. We, therefore, used the RNA probe to detect the presence of non-coding human DNA in NIH3T3 cells treated with cfChPs isolated from human serum (*Figure 17b*). We discovered that the internalised cfChPs concatemers reacted with probes against both human whole genomic DNA and human non-coding DNA, and their fluorescent signals co-localised to the extent of 98.2% (*Figure 17b*). This finding was confirmed on metaphase preparations of the cfChPs treated cells (*Figure 17c*). This led us to the conclusion that virtually all the cfChPs concatemers were composed of non-coding DNA.

## Biological activities of concatemers are largely attributable to non-coding DNA

The above finding that the concatemers are largely composed of non-coding DNA led us to investigate whether the biological activities of the concatemers that we described earlier were attributable to non-coding DNA. We discovered this to be indeed the case. We found that the fluorescent signals of a selected set of target enzymes, viz. DNA polymerase, RNA polymerase, and reverse transcriptase co-localised with those of non-coding DNA, as did BrdU signals representing DNA synthesis

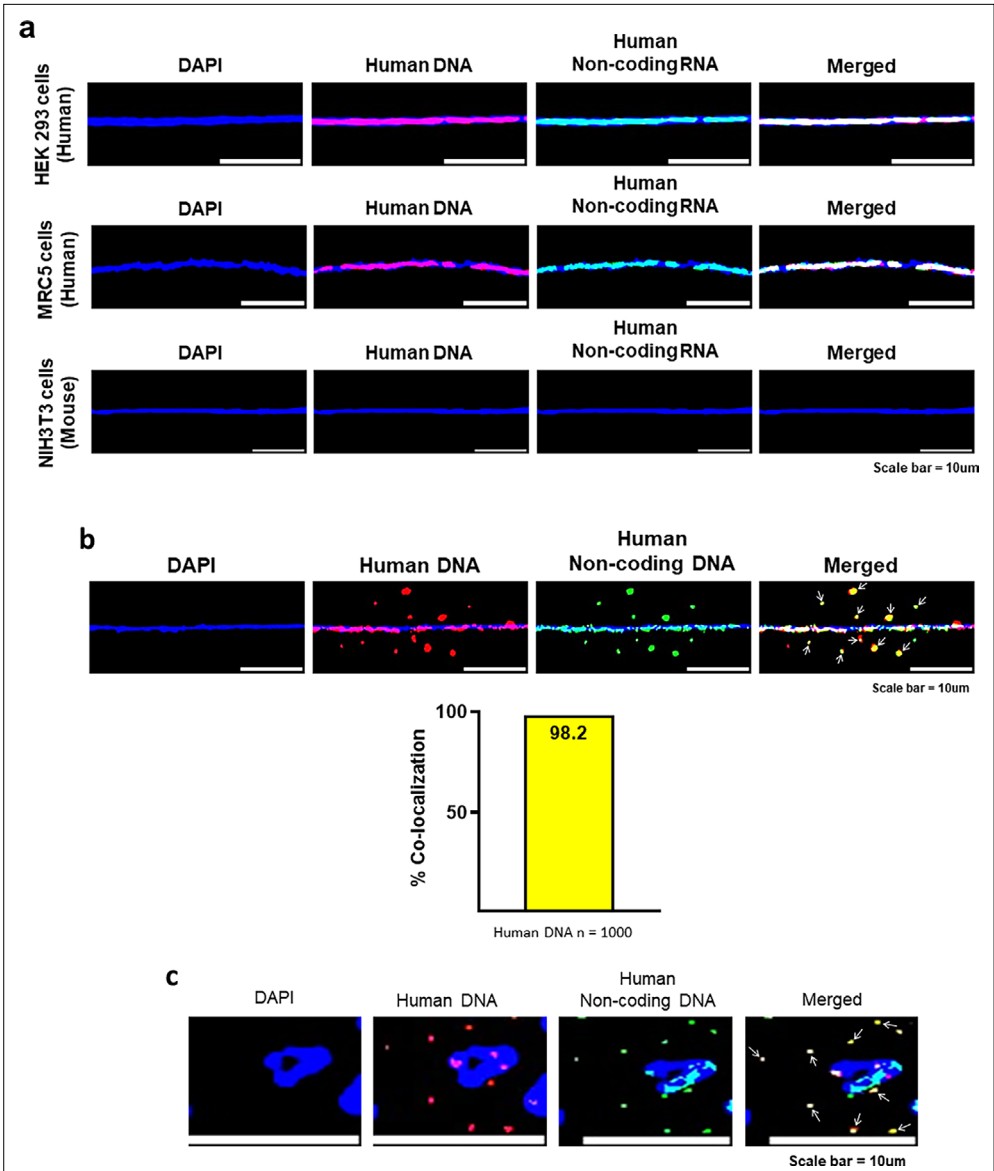

**Figure 17.** The concatemers are largely composed of non-coding DNA. (**a**) The human long non-coding RNA probe aligns with human DNA but not with mouse DNA. Chromatin fibres were prepared from two human cell lines viz. HEK293, MRC5 and one mouse cell line viz. NIH3T3 and the cells were hybridised with the long non-coding RNA probe. The results show that the probe has almost 100% coverage in case of the two human cells but does not react with the mouse cells. (**b**) The concatemers are largely composed of non-coding DNA. cfChPs treated passaged cells were simultaneously probed with a fluorescently labelled human whole genomic DNA FISH probe and the human long non-coding RNA probe (which substituted for a non-coding DNA probe). The degree of co-localisation of fluorescent signals is represented as a histogram, which was generated after counting 1000 human DNA signals, and which shows that the extent of co-localisation of the fluorescent signals was of the order of 98.2%. (**c**) Confirmation of these findings on metaphase spreads.

(*Figure 18*). As anticipated, the fluorescence signals of LINE-1 and Alu also co-localised with those of non-coding DNA (*Figure 18*). The degree of co-localisation of fluorescent signals of non-coding DNA and those of the above biomarkers is given in *Figure 18—figure supplement 1*. The latter also shows that the degree of co-localisation with the biomarkers was similar irrespective of whether we used a whole human genomic probe (results of which are shown earlier) or the one against non-coding DNA. Collectively, these findings provided confirmation that the biological activities of the concatemers are attributable to non-coding DNA and suggested that non-coding DNA has many unique biological

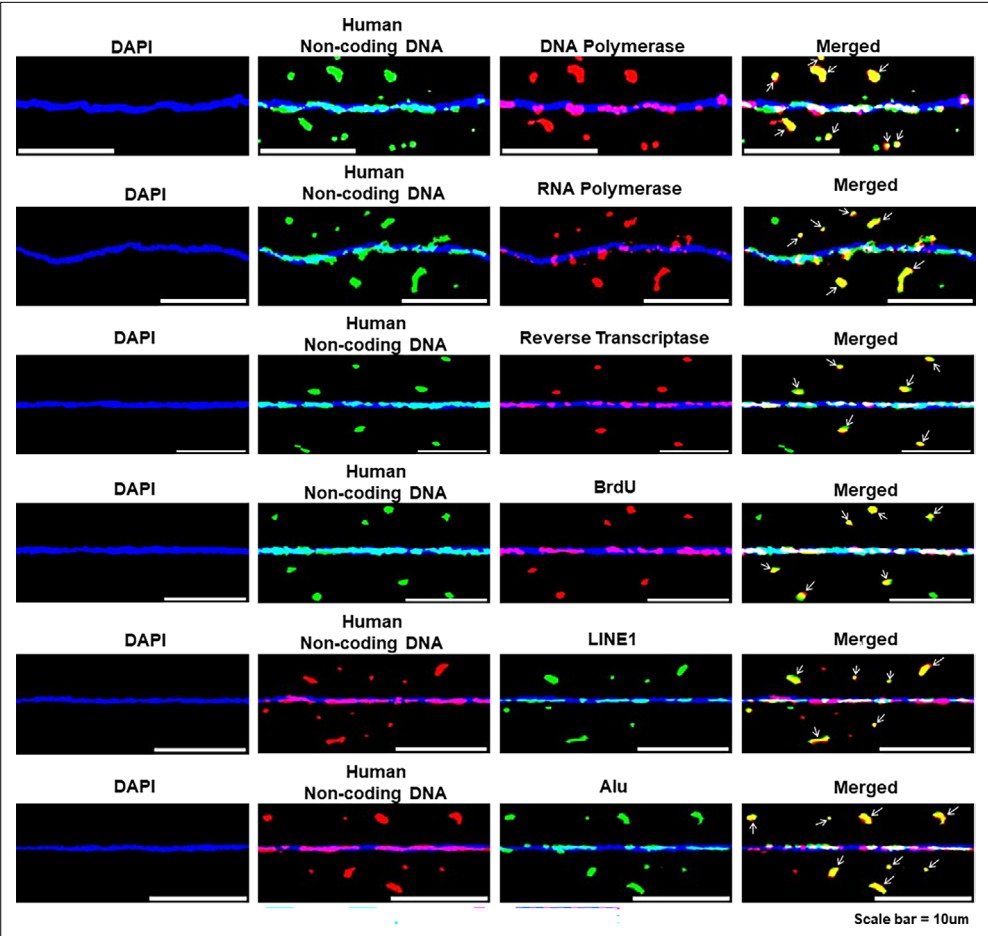

**Figure 18.** Biological activities of the concatemers are attributable to non-coding DNA. Immuno-FISH and dual-FISH analysis of chromatin fibres prepared from cfChPs-treated cells in continuous passage. The chromatin fibres were probed with antibodies against various human proteins and the human non-coding RNA probe (which substituted for a non-coding DNA probe) or the human non-coding RNA probe and FISH probes against LINE-1 and Alu. Co-localisation of various signals is clearly seen. These data suggested that biological activities of the concatemers are attributable to non-coding DNA.

The online version of this article includes the following figure supplement(s) for figure 18:

**Figure supplement 1.** Biological activities of the concatemers are attributable to non-coding DNA.

functions which remain dormant but are activated following cellular apoptosis to become detectable in association with the cfChP concatemers.

## Experiments using cfChPs isolated from healthy individuals

The experiments described above were performed with NIH3T3 cells that had been treated with cfChPs isolated from the sera of cancer patients and passaged continuously. We simultaneously performed similar experiments with NIH3T3 cells that had been treated with cfChPs isolated from the sera of healthy individuals. We found that cfChPs from healthy individuals were also readily internalised by NIH3T3 cells, wherein they behaved in a manner similar to that of cfChPs derived from the cancer patients (*Figure 19*).

## Experiments using cfChPs released from hypoxia-induced dying MDA-MB-231 cells

Finally, given our previous finding that cfChPs spontaneously released from dying cancer cells were readily internalised by healthy cells (*Mittra et al., 2017*), we further investigated the intracellular activities of cfChPs released from dying MDA-MB-231 breast cancer cells. Hypoxia-induced cfChPs were

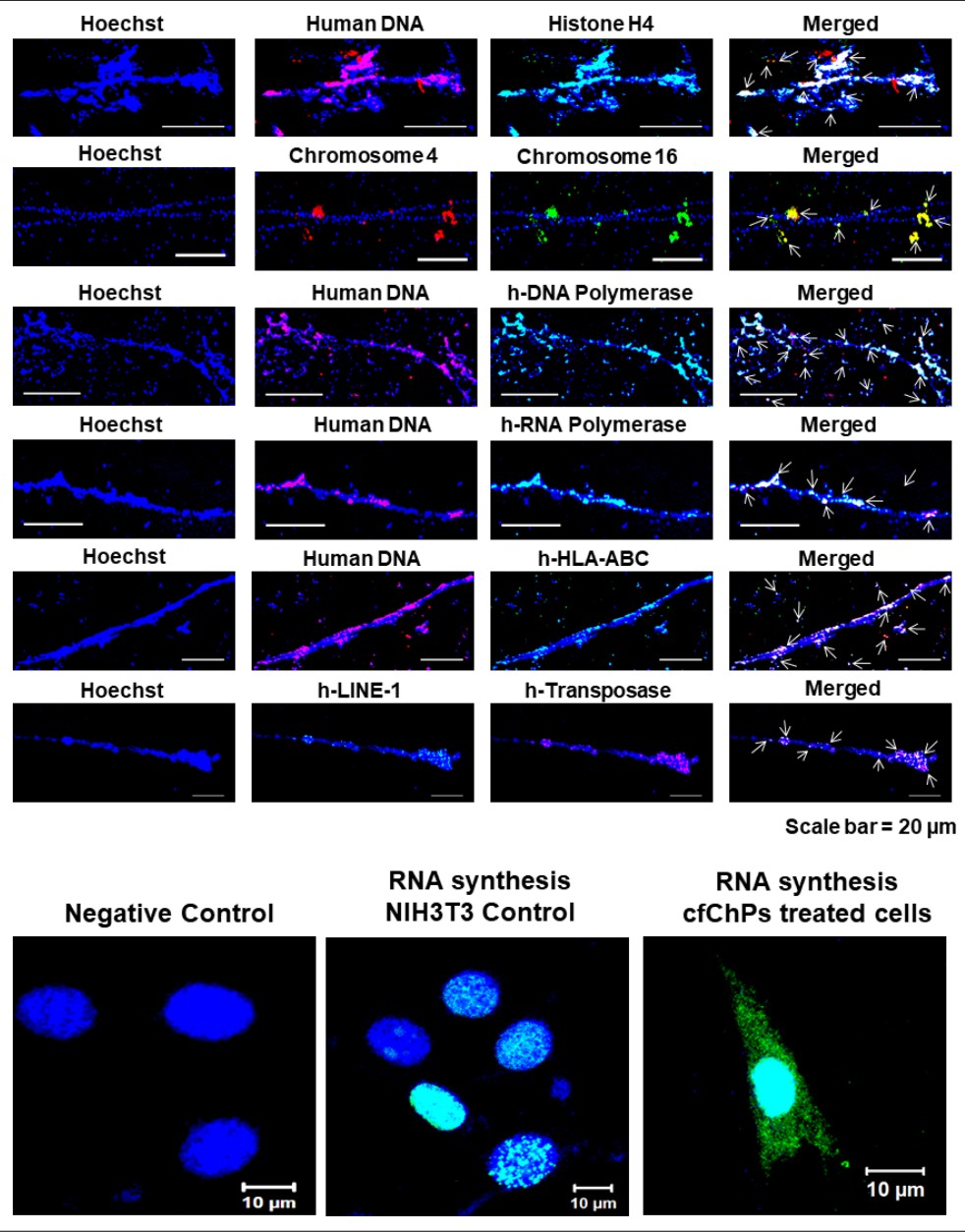

**Figure 19.** NIH3T3 cells treated with cfChPs isolated from the serum of healthy individuals and which were kept in continuous passage show similar characteristics and properties as those of cfChPs isolated from cancer patients.

collected in 1 ml of culture medium and added to NIH3T3 cells (please see Materials and methods). We observed that cfChPs that were released from hypoxia-induced dying MDA-MB-231 cells could reciprocate all the intracellular activities and functions that were observed using cfChPs isolated from serum of both cancer patients and healthy individuals (*Figure 20*).

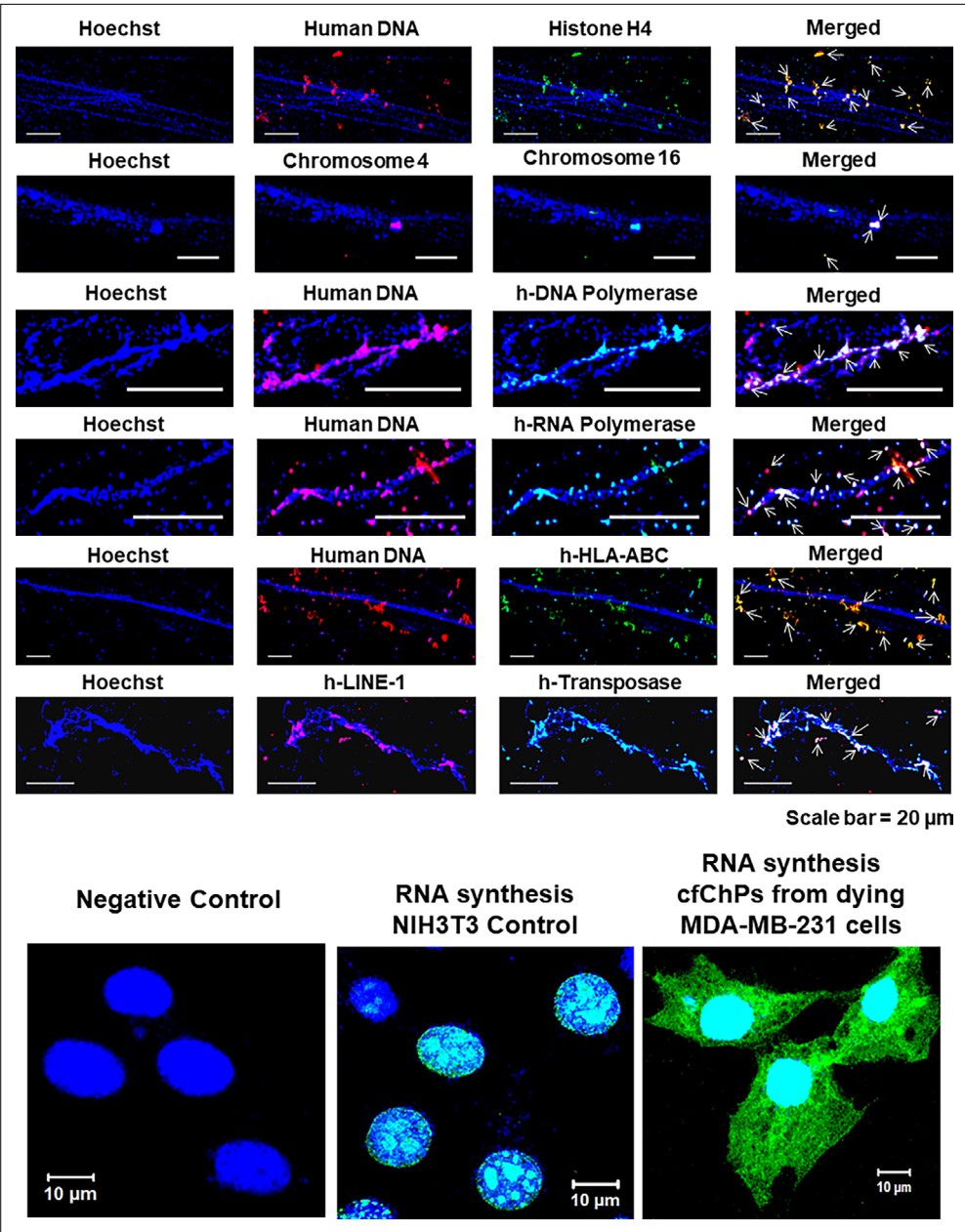

**Figure 20.** NIH3T3 cells treated with conditioned medium containing cfChPs released from dying MDA-MB-231 breast cancer cells and which were kept in continuous passage show similar characteristics and properties as those of cfChPs isolated from cancer patients and healthy individuals.

## Discussion

The present study is founded on our earlier reports that cfChPs released from the billions of cells that die in the body every day can horizontally transfer themselves to healthy cells resulting in activation of several biological processes (*Mittra et al., 2015b*; *Mittra et al., 2017*). Herein, we have reported multiple additional activities and biological processes that can be summarised as follows. The internalised cfChPs containing widely disparate DNA sequences randomly combined to form complex concatemers of variable sizes, many of which were ostensibly multi-mega base pairs in size. The concatemers exhibited variable and bizarre spatial relationships with the host cell interphase DNA, with many remaining in the cytoplasm and others being directly associated with the mouse chromosomal DNA. The concatemers once formed persisted across cell generations and showed remarkable stability. They exhibited the potential to perform many functions that are normally attributed

to the nuclear genome. They could generate DNA polymerase which allowed them to synthesise DNA and independently replicate themselves without heed to the mitotic cycle of the host cell. The concatemers synthesised RNA as well as RNA polymerase, ribosomal RNA, ribosomal proteins, and numerous other human proteins in the mouse cells. The latter manifested as complex multi-peptide fusion proteins that were often highly overexpressed. Further analysis demonstrated that the concatemers harboured human LINE-1 and *Alu* retro-transposable elements which were associated with the enzyme DNA polymerase. The latter allowed them to increase their copy numbers and extensively amplify themselves with time in culture. The LINE-1 and Alu elements were also associated with reverse transcriptase and transposase, suggesting that they had the potential to rearrange themselves within the mouse genome leading to extensive modifications. Taken together, the above findings lead us to conclude that the concatemers that form following horizontal transfer of cfChPs act as autonomous satellite genomes which function independently of the nuclear genome. Being associated with transposable elements, they have the potential to modify the host genome in unpredictable ways, leading to somatic mosaicism - the hallmark of ageing and cancer (*Vijg, 2014*). Finally, the above findings could be reproduced in four other cell lines derived from different species, suggesting that horizontal transfer of cfChPs may be a universal phenomenon.

Some of the summary points mentioned above call for further elaboration. We discovered that the concatemers were primarily composed of non-coding DNA. The latter is not normally known to perform the diverse biological activities that we detected to be associated with the concatemers. This suggested that non-coding DNA has many hidden biological functions which remain dormant but are activated following cellular apoptosis to become detectable in association with the cfChP cocatemers. Our demonstration that the concatemers synthesise ribosomal RNA, RNA polymerase, and ribosomal protein suggests that they are capable of creating their own autonomous protein synthetic machinery without recourse to that of the host cell. The concatemers being composed of open chromatin are apparently able to indiscriminately transcribe their DNA via RNA polymerase, and the RNAs thus generated are translated into proteins with the help of their own ribosomal RNA and ribosomal proteins. Significantly, the newly synthesised proteins remained strictly associated with their concatemers of origin, suggesting that, although the concatemers contained the critical components of a protein synthetic machinery, they lacked the machinery required for protein sorting. Taken together, these findings suggest a novel mechanism of protein synthesis which functions autonomously without recourse to the host cell's own protein synthetic machinery. Clearly, such an assertion, which defies existing knowledge, requires critical experimental scrutiny to define the mechanism(s) that underlie such a unique biological process.

Our detection of the unusual intracellular characteristics and functions of cfChPs and their concatemers was facilitated by several factors. First, our trans-species model comprising mouse recipient cells and cfChPs derived from human serum. This allowed us to track the activities of internalised human cfChP concatemers within the mouse cells using appropriate human-specific FISH probes and antibodies. Second, the chromatin fibre technique facilitated the detection of spatial relationships between internalised cfChPs concatemers and the mouse interphase DNA. Third, our use of human-specific DNA FISH probes and antibodies allowed the detection of short DNA sequences within the concatemers and small peptides that they synthesised by binding to the corresponding DNA sequences and peptide epitopes, respectively. These would not have been readily detectable by the standard techniques of molecular biology (discussed subsequently in the manuscript). Finally, our discovery that the non-coding RNA probe could be substituted to detect non-coding DNA allowed us to confirm that the concatemers were primarily composed of non-coding DNA and that they could perform many of the functions that are normally attributed to the genomic DNA.

Much research has been devoted to transposable elements since the seminal discovery of Barbara McClintock of 'jumping genes' in maize. The sequencing of the human genome has revealed that approximately 40% of the genome is composed of retro-transposable elements (*Lander et al., 2001*), and the latter has been shown to numerically increase with age (*De Cecco et al., 2013*). Although many theories have been proposed for this age-related increase (*Yushkova and Moskalev, 2023*), our results would suggest that the concatemers containing LINE-1 and Alu elements proliferate and amplify with time in culture by virtue of their association with DNA polymerase. Furthermore, the association of LINE-1 and Alu elements within the concatemers with reverse transcriptase and transposase suggested that the amplified retro-transposable elements have the capability to indiscriminately

rearrange the host genome, potentially resulting in a degree of somatic mosaicism that would prevent the assembly of a contiguous genome. We found that this may indeed be the case, as we observed that the cfChPs-treated NIH3T3 cells began to die out beyond passage number ~250. It should be mentioned in this context that retro-transposable elements are implicated in cancer due to their ability to create genomic structural variation, alterations, and instability (*Burns, 2017*). We have earlier hypothesised that illegitimate and repeated genomic integration of cfChPs may be the underlying aetiological factor for cancer as well as ageing and age-related chronic diseases (*Raghuram et al., 2019*). Taken together, the above discussion suggests that horizontal transfer of cfChPs has profound effects on genome structure and function that are critical to human health and disease. The results also raise the possibility of developing potential therapies for prevention of cancer and retardation of ageing using agents that could deactivate cfChPs before they entered into new host cells (*Mittra, 2024*; *Khanvilkar and Mittra, 2025*; *Pal et al., 2022*; *Pilankar et al., 2022*; *Bandiwadekar et al., 2025*).

Transposable elements carried by the concatemers may also play a significant role in mammalian evolution by inserting themselves into unpredictable locations, disrupting or restoring gene function, generating new regulatory elements, and potentially inducing genomic rearrangements and instability (*Senft and Macfarlan, 2021*; *Pourrajab and Hekmatimoghaddam, 2021*). However, the origin of transposable elements has remained in the realm of speculation. Many transposable elements are thought to represent remnants of ancient viruses that had become integrated into host genomes millions of years earlier (*Gilbert and Feschotte, 2018*). Our demonstration that the concatemers carry LINE-1 and Alu elements suggests transposable elements are 'foreign' genetic elements that are acquired from dying cells via HGT. It has been proposed that generation of novel proteins resulting from gene duplication, sequence divergence, and gene combination may also have evolutionary functions (*Bashton and Chothia, 2007*; *Zhang et al., 2019*). We speculate that the plethora of novel fusion proteins that we discovered may likewise play a role in the evolutionary process.

Conventionally, HGT refers to the transfer of genetic material from one organism to another. Although HGT occurs extensively in microorganisms, it has thus far been difficult to detect HGT in mammals. Our results suggest that HGT occurs in mammalian cells via cfChPs released from the billions of host cells that die on a daily basis. We previously proposed that circulating cfChPs released from dying cells function as a new class of mobile genetic elements (MGEs) that act intra-corporeally and transfer themselves horizontally to the organism's own cells (*Mittra, 2015a*). Thus, 'within-self' HGT may occur in mammals on a massive scale via the medium of cfChP concatemers that have undergone extensive and complex modifications resulting in their behaviour as 'foreign' genetic elements. However, for concatemers to have evolutionary relevance in mammals, they need to be horizontally transferred to the germ line. The latter is not unlikely given the high turnover of daily cell death and release of cfChPs in the blood circulation which could carry them to all cells and tissues of the body, including the germ line cells.

Numerous attempts on our part to characterise the concatemers by conventional real-time PCR were unsuccessful. Given the complex and chaotic amalgams of disparate DNA fragments that comprise the concatemers, the primer sequences specific to a particular gene could not find the appropriate DNA sequences to bind to. Advanced sequencing techniques may help to define the complex nature of the DNA sequences that comprise the concatemers. Likewise, multiple attempts to detect the components of the fusion proteins by western blotting also failed. Even if an antibody reacted to a peptide component of a fusion protein, the band obtained would not correspond to the molecular weight of the positive control band, and the possibility of it being an artefact could not be excluded. Studies using liquid chromatography with tandem mass spectrometry may help to delineate the composition of the fusion proteins.

Nevertheless, it is worth mentioning here that we have earlier reported results of whole genome sequencing of NIH3T3 cells that had been treated with cfChPs isolated from serum of cancer patients under identical conditions as used in the present study (*Mittra et al., 2015b*). Whole genome sequencing on Illumina GA IIX platform and bioinformatics analysis of two single cell clones (D5 and B2) developed from the cfChPs treated cells detected 25,000–28,000 strictly human reads in the mouse cell genomes. It is likely that our highly stringent alignment criteria had greatly underestimated the number of human sequences that we detected the mouse cells in that study. Computational analysis of the two clones also detected 35–47 unique human Alu repeat sequences in the recipient cells

which were confirmed by PCR and Sanger sequencing. Taken together, these data support our current results that human DNA in the form of cfChPs is present in large numbers in the treated mouse cells and carry Alu elements within them.

The presence of extrachromosomal DNA (ecDNA) has been described in plants (*Wong and Wildman, 1972*), yeast (*Møller et al., 2015*), *Caenorhabditis elegans* (*Shoura et al., 2017*), *Drosophila* (*Stanfield and Lengyel, 1979*), mice, and humans (*Møller et al., 2018*; *Shibata et al., 2012*). Recently, there has been a resurgence of interest in the role of ecDNA in cancer (*Wu et al., 2022*; *Yan et al., 2024*). In particular, it has been suggested that ecDNA can be a source of extreme gene amplification enabling a cell to harbour multiple copies of oncogenes and can potentially promote tumour heterogeneity and therapy resistance (*Wu et al., 2022*; *Yan et al., 2024*). It is believed that ecDNAs are derived from the host cell's own genome resulting from chromothripsis, DNA damage, or genomic instability (*Wu et al., 2022*). Our study, which also detected multiple highly amplified oncogene products, often appearing as fusion proteins, raises the possibility that ecDNAs that have been detected in human cancers may represent concatemers composed of cfChPs acquired from surrounding dying cancer cells via HGT. The origin and biological properties of ecDNA call for further in-depth investigations in light of our current findings.

Our study leaves many unanswered questions, and our findings will require critical validation in future research. As such, our results should be viewed as representing an expansive theory, wherein the activities of the concatemers formed by amalgamation of horizontally transferred cfChPs are governed by unknown biological processes that are distinct from those that govern the nuclear genome. Our results also suggest that non-coding DNA may have many unique biological functions which remain dormant but are activated following cellular apoptosis to become detectable in association with the cfChP cocatemers. An understanding of these unexplored biological processes could offer a new perspective on the process of mammalian evolution.

In summary, the present findings lead to a novel hypothesis that a cell simultaneously harbours two genome forms that function independently of each other: one that is inherited (hereditary genome) and numerous others that are acquired (satellite genomes). The latter, acting as autonomous genomic agents, blur the line between inherited and acquired genetic material, opening new research avenues in biology and medicine. The satellite genomes may potentially perform evolutionary functions by acting as vehicles of transposable elements and generating a plethora of novel proteins. Our findings suggest that the biological life of the genome is not a co-terminus with the death of a cell, but rather that the genome is recycled in a fragmented and extensively modified incarnation to perform new functions in distant destinations acting as 'within-self' foreign genetic elements. This suggests that a cell's propensity to internalise cfChPs represents a natural biological process designed to modify the hereditary genome, which in turn may promote evolutionary change. We hypothesise that mammalian evolution is driven by 'within-self' HGT of cfChP concatemers that have undergone such extensive and complex modifications that they behave as 'foreign' genetic elements.

## Materials and methods
### Isolation of cfChPs from human serum
cfChPs were isolated from sera of five patients suffering from cancer and five healthy volunteers using the protocol described by us earlier (*Mittra et al., 2015b*). Serum from five donors of each group was pooled prior to a cfChPs isolation. The steps of the isolation protocol can be summarised as follows: (1) centrifugation of the pooled serum samples (1 mL) at 700,000 × *g* for 16 hr at 4 °C; (2) treating the pellet obtained with lysis buffer; (3) centrifugation of the lysate at 700,000 × *g* for 16 hr at 4 °C; (4) suspension of the pellet in 1 mL PBS; (5) passing the suspension through an affinity column (Thermo Fisher Scientific, USA) containing biotinylated anti-histone H4 antibody (125 µg) bound to 2 mL of Pierce Streptavidin Plus Ultralink Resin (Thermo Fisher Scientific, USA); (6) elution of the column with 1 mL 0.25 M NaCl; (7) ultra-centrifuging the elute as described above and suspending the pellet containing cfChPs in 1 mL PBS. The presence of cfChPs was confirmed using a nucleosome specific sandwich ELISA kit (Cell Death Detection ELISA^PLUS kit, Roche Diagnostics GmbH, Germany). A representative electron microscopy image of the isolated cfChPs is given in *Figure 1*. The concentrations of cfChPs in the isolates are expressed in terms of their DNA content, as estimated using the PicoGreen

dsDNA quantitation assay (Thermo Fisher Scientific, USA; *Mittra et al., 2015b*). The age, sex, and tumour types of the participants are given in *Supplementary file 2*.

## Cell lines and culture

A list of the various cell lines used in this study, their tissue origin and procurement source, is given in *Supplementary file 3*. Origin of all cell lines was authenticated by species-specific DNA FISH probes. NIH3T3 and MDA-MB-231 cells were additionally authenticated by PCR using species-specific primers of housekeeping genes. Mycoplasma negativity testing was done using the Mycoplasma antigen detection method and by PCR. NIH3T3 mouse fibroblast cells were maintained in Dulbecco's modified Eagle medium (DMEM) containing 10% bovine calf serum and grown at 37 °C in 35 mm culture dishes in an atmosphere of 95% air, 5% $CO_2$, and 95% humidity. The cells ($10 \times 10^4$) were treated with 10 ng of cfChPs isolated from the serum of cancer patients or healthy individuals and maintained in continuous culture by passaging every fourth day. We chose to use 10 ng based on our earlier report in which we had obtained robust biological effects such as activation of DDR and apoptotic pathways using this concentration of cfChPs (*Mittra, 2015a*).

## Collection of conditioned medium containing cfChPs released from dying MDA-MB-231 human breast cancer cells

In addition to cfChPs isolated from the cancer patients and healthy individuals, we also investigated the effects of treating NIH3T3 cells with cfChPs that were released into the conditioned medium from hypoxia-induced dying MDA-MB-231 human breast cancer cells. The protocol for collecting cfChPs from hypoxia-induced dying cells has been described by us in detail earlier (*Raghuram et al., 2024*). The only difference was that in the earlier study, we had used dying NIH3T3 cells instead of MDA-MB-231 cells for collecting media containing hypoxia-induced cfChPs. In the dual-chamber system used in these experiments, the pore size of the filter separating the two chambers was 400 μm. Consequently, the cfChPs from the dying MDA-MB-231 cells that were collected in the lower chamber were <400 μm in size. NIH3T3 cells ($1 \times 10^5$) were treated with 1 ml of the hypoxic media containing cfChPs.

## Fluorescent dual labelling of cfChPs

In some experiments, the DNA and histones of the isolated cfChPs from serum were fluorescently dually labelled with Platinum Bright 550 Red Nucleic Acid Labelling Kit (Kreatech Diagnostics, Cat # GLK-004) and ATTO 488 NHS-ester (ATTO-TEC GmbH, Cat # AD488-35), respectively, according to our previously reported protocol (*Mittra et al., 2015b*). NIH3T3 cells were treated with 10 ng of dually labelled cfChPs for 6 hr, washed with PBS x 3, and examined under a confocal microscope (Carl-Zeiss, GmbH, Germany). A representative image of dually labelled cfChPs is given in *Figure 2a*.

## Preparation of chromatin fibres

Chromatin fibres were prepared from NIH3T3 cells according to previously described methods (*Nieminuszczy et al., 2016*; *Quinet et al., 2017*) with minor modifications. Although the inventors of this technique had described the fibres as DNA fibres, when stained with histone H4 antibody, we found the fibres to comprise chromatin (*Figure 21*). To prepare the chromatin fibres, 2 μLl of the cfChPs-treated NIH3T3 cell suspension was spotted at one end of a glass slide and semi-evaporated for 10 min at room temperature (~25 °C). The spotted cells were incubated with 7 μL of lysis buffer (0.1% sodium dodecyl sulphate in phosphate-buffered saline) for 2 min with gentle shaking, and the slides were tilted at an angle of 15°–25° to allow the cell lysate containing the chromatin fibres to roll down along the slide surface. The slides were fixed with chilled methanol for 10 min and processed for immunofluorescence and FISH analyses.

## Metaphase spread preparation

Metaphase spreads from NIH3T3 cells treated with cfChPs in continuous culture were prepared using a standard protocol (*Mittra et al., 2015b*). The slides were processed for immunofluorescence and FISH analyses.

## Human specificity of antibodies and FISH probes

Sources and other details of the antibodies and FISH probes used in this study are provided in *Supplementary file 1*. Irrespective of the specifications of the antibodies given in the vendors' data sheets,

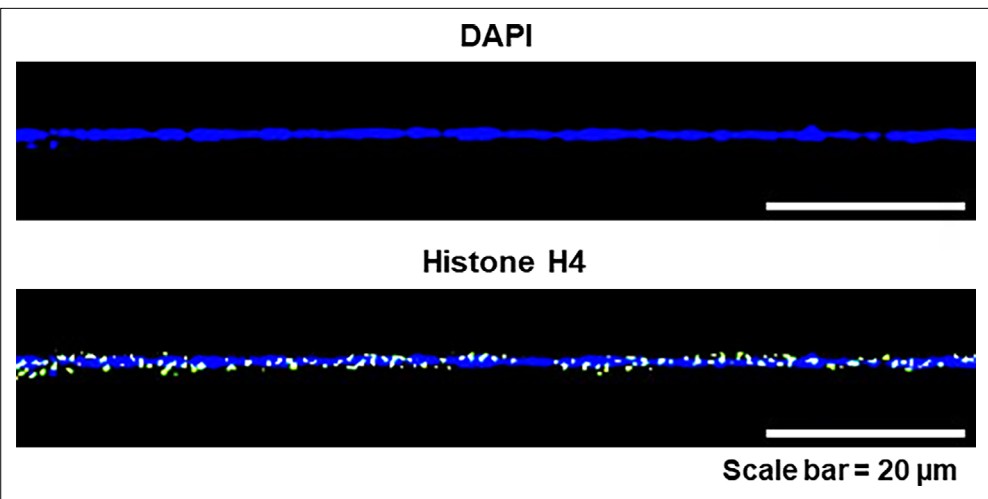

**Figure 21.** Representative image to confirm that fibres prepared from NIH3T3 cells are chromatin fibres and not DNA fibres. The fibres were stained with DAPI and anti-histone H4 antibody.

and the custom synthesised FISH probes, we independently verified their human specificity by undertaking extensive positive and negative control experiments to ensure that all antibodies and FISH probes were human specific. The results of these control experiments are given in *Figure 3—figure supplement 1*, *Figure 5—figure supplement 1*, *Figure 9—figure supplement 1*, *Figure 10—figure supplement 3*, *Figure 11—figure supplement 2*, *Figure 12—figure supplement 1*, *Figure 4—figure supplement 1*.

## Immunofluorescence and FISH

The expression of various proteins in the cells was evaluated using indirect immunofluorescence, and the presence of human DNA in chromatin fibres and metaphase spreads was detected using FISH, as described previously (*Mittra et al., 2015b*). Briefly, for immunofluorescence, the chromatin fibres were fixed in 4% paraformaldehyde, blocking in saponin buffer containing 10% normal goat serum followed by immune-staining with appropriate primary and secondary antibodies. In order to remove unbound antibodies, 1 X PBS containing 0.05% Tween20 washes were given x 3 before mounting with VectaShield DAPI for microscopic examination. In case of immunofluorescence on metaphase spreads, the paraformaldehyde fixation step was omitted. For FISH on both chromatin fibres and metaphase spreads, the slides were dehydrated using alcohol series (70%, 80%, and 100%) followed by hybridisation overnight with appropriate FISH probes in a humidified chamber at 37 °C. In order to remove the unbound probes, the slides were washed in 0.4 X SSC at 70 °C for 1 min followed by 4 X SSCT (4 X SSC in 0.05% Tween20) washes at 45 °C for 5 min each. The final wash was in 4 X SSCT for 2 minutes at room temperature followed by mounting with VectaShield DAPI for microscopic examination. For immuno-FISH experiments, the slides were first stained for immunofluorescence as described above and fixed with 2% paraformaldehyde for 10 min followed by FISH procedure as described above. For microscopic examination, immunofluorescence slides were imaged at ×400 magnification while FISH slides were analysed at ×600 under an Applied Spectral Bio-imaging System (Applied Spectral Imaging, Israel). The immuno-FISH slides were analysed at ×600 magnification.

## Detection of DNA synthesis

To detect DNA synthesis on chromatin fibres and metaphase spreads, cells were pulse-labelled while in culture with 10 μM bromodeoxyuridine (BrdU) for 24 hr, washed in PBS x 3, trypsinised and used for preparing chromatin fibres and metaphase spreads. Newly synthesised DNA was detected using anti-BrdU antibody.

## Detection of RNA synthesis

RNA synthesis in cfChP-treated cells in continuous passage was detected using an Abcam assay kit (Cambridge, UK; Catalogue No. ab228561) as per the manufacturer's protocol. With the use

of metabolic tagging with 5-ethynyluridine (EU), this assay offers a reliable method for fluorescent staining of newly produced RNA with the click chemistry facilitating viewing. In some experiments, the cfChPs-treated cells were treated with actinomycin D (0.0005 µg/mL) for 24 hr or grown at low temperature (31 °C) for 24 hr.

## Statistical analysis

Statistical analysis was done by Student's t-test and by ANOVA for linear trend. Both analyses were done using GraphPad version 8 (RRID:SCR_002798).

## Acknowledgements

We thank Mr. Ashish Pawar for his help in preparing this manuscript. We would also like to thank Editage (https://www.editage.com/) for English language editing support. This study was supported by the Department of Atomic Energy, Government of India, through its grant CTCTMC to the Tata Memorial Centre awarded to IM. The funding agency had no role in research design, collection, analysis, and interpretation of data, or manuscript writing.

## Additional information

### Funding

| Funder | Grant reference number | Author |
|---|---|---|
| Department of Atomic Energy, Government of India | CTC-TMC | Indraneel Mittra |

The funders had no role in study design, data collection and interpretation, or the decision to submit the work for publication.

### Author contributions

Soumita Banerjee, Soniya Sanjay Shende, Laxmi Kata, Relestina Simon Lopes, Swathika Praveen, Ruchi Joshi, Naveen Kumar Khare, Methodology; Gorantla V Raghuram, Snehal Shabrish, Formal analysis, Supervision, Writing – original draft; Indraneel Mittra, Conceptualization, Formal analysis, Supervision, Funding acquisition, Writing – original draft, Project administration, Writing – review and editing

### Author ORCIDs

Soumita Banerjee ⓘ https://orcid.org/0000-0002-5909-9880
Relestina Simon Lopes ⓘ https://orcid.org/0009-0004-9615-6700
Swathika Praveen ⓘ https://orcid.org/0000-0001-9926-2323
Naveen Kumar Khare ⓘ https://orcid.org/0000-0002-6025-3785
Gorantla V Raghuram ⓘ https://orcid.org/0000-0001-5132-4249
Snehal Shabrish ⓘ https://orcid.org/0000-0002-6407-5082
Indraneel Mittra ⓘ https://orcid.org/0000-0002-5768-3821

### Ethics

This study was approved by the Institutional Ethics Committee (IEC) of the Advanced Centre for Treatment, Research and Education in Cancer, Tata Memorial Centre for the collection of blood (10 mL) from cancer patients and healthy individuals for the isolation of cfChPs (approval no. 900520). A formal informed consent form that was approved by the IEC was signed by each participant.

Reviewer #1 (Public review): https://doi.org/10.7554/eLife.103771.3.sa1
Author response https://doi.org/10.7554/eLife.103771.3.sa2

# Additional files

## Supplementary files
MDAR checklist

Supplementary file 1. Antibodies and FISH probes used in this study (custom synthesized as per vendors' specifications).

Supplementary file 2. Clinical and demographic information of cancer patients and healthy individuals who provided blood samples for isolation of cell-free chromatin particles.

Supplementary file 3. Cell lines used in this study.

## Data availability
All data generated during this study are included in this published article and supplementary information.

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
