## [Editor Report · eLife Assessment]

The authors examine the effect of cell-free chromatin particles (cfChPs) derived from human serum or from dying human cells on mouse cells in culture and propose that these cfChPs can serve as vehicles for cell-to-cell active transfer of foreign genetic elements. The work presented in this paper is intriguing and potentially **important**, but it is **incomplete**. At this stage, the claim that horizontal gene transfer can occur via cfChPs is not well supported because it is only based on evidence from one type of methodological approach (immunofluorescence and fluorescent in situ hybridization (FISH)) and is not validated by whole genome sequencing.

---

## [Referee Report · Reviewer #1 (Public review)]

Summary:

Horizontal gene transfer is the transmission of genetic material between organisms through ways other than reproduction. Frequent in prokaryotes, this mode of genetic exchange is scarcer in eukaryotes, especially in multicellular eukaryotes. Furthermore, the mechanisms involved in eukaryotic HGT are unknown. This article by Banerjee et al. claims that HGT occurs massively between cells of multicellular organisms. According to this study, the cell free chromatin particles (cfChPs) that are massively released by dying cells are incorporated in the nucleus of neighboring cells. These cfChPs are frequently rearranged and amplified to form concatemers, they are made of open chromatin, expressed, and capable of producing proteins. Furthermore, the study also suggests that cfChPs transmit transposable elements (TEs) between cells on a regular basis, and that these TEs can transpose, multiply, and invade receiving cells. These conclusions are based on a series of experiments consisting in releasing cfChPs isolated from various human sera into the culture medium of mouse cells, and using FISH and immunofluorescence to monitor the state and fate of cfChPs after several passages of the mouse cell line.

Strengths:

The results presented in this study are interesting because they may reveal unsuspected properties of some cell types that may be able to internalize free-circulating chromatin, leading to its chromosomal incorporation, expression, and unleashing of TEs. The authors propose that this phenomenon may have profound impacts in terms of diseases and genome evolution. They even suggest that this could occur in germ cells, leading to within-organism HGT with long-term consequences.

Weaknesses:

The claims of massive HGT between cells through internalization of cfChPs are not well supported because they are only based on evidence from one type of methodological approach: immunofluorescence and fluorescent in situ hybridization (FISH) using protein antibodies and DNA probes. Yet, such strong claims require validation by at least one, but preferably multiple, additional orthogonal approaches. This includes, for example, whole genome sequencing (to validate concatemerization, integration in receiving cells, transposition in receiving cells), RNA-seq (to validate expression), ChiP-seq (to validate chromatin state).

Should HGT through internalization of circulating chromatin occur on a massive scale, as claimed in this study, and as illustrated by the many FISH foci observed on Fig 3 for example, one would expect that the level of somatic mosaicism may be so high that it would prevent assembling a contiguous genome for a given organism. Yet, telomere-to-telomere genomes have been produced for many eukaryote species, calling into question the conclusions of this study.

---

## [Author Response]

[The following is the authors’ response to the current reviews.]

**eLife Assessment**
The authors examine the effect of cell-free chromatin particles (cfChPs) derived from human serum or from dying human cells on mouse cells in culture and propose that these cfChPs can serve as vehicles for cell-to-cell active transfer of foreign genetic elements. The work presented in this paper is intriguing and potentially important, but it is incomplete. At this stage, the claim that horizontal gene transfer can occur via cfChPs is not well supported because it is only based on evidence from one type of methodological approach (immunofluorescence and fluorescent in situ hybridization (FISH)) and is not validated by whole genome sequencing.

We disagree with the eLife assessment that our study is incomplete because we did not perform whole genome sequencing. Tens of thousands of genomes have been sequenced, and yet they have failed to detect the presence of the numerous “satellite genomes” that we describe in our paper. To that extent whole genome sequencing has proved to be an inappropriate technology. Rather, eLife should have commended us for the numerous control experiments that we have done to ensure that our FISH probes and antibodies are target specific and do not cross-react.

**Public Reviews:**

**Reviewer #1 (Public review):**
Summary:Horizontal gene transfer is the transmission of genetic material between organisms through ways other than reproduction. Frequent in prokaryotes, this mode of genetic exchange is scarcer in eukaryotes, especially in multicellular eukaryotes. Furthermore, the mechanisms involved in eukaryotic HGT are unknown. This article by Banerjee et al. claims that HGT occurs massively between cells of multicellular organisms. According to this study, the cell free chromatin particles (cfChPs) that are massively released by dying cells are incorporated in the nucleus of neighboring cells.

The reviewer is mistaken. We do not claim that the internalized cfChPs are incorporated into the nucleus. We show throughout the paper that the cfChPs perform their novel functions autonomously outside the genome without being incorporated into the nucleus. This is clearly seen in all our chromatin fibre images, metaphase spreads and our video abstract. Occasionally, when the cfChPs fluorescent signal overlie the chromosomes, we have been careful to state that the cfChPs are associated with the chromosomes without implying that they have integrated.

These cfChPs are frequently rearranged and amplified to form concatemers, they are made of open chromatin, expressed, and capable of producing proteins. Furthermore, the study also suggests that cfChPs transmit transposable elements (TEs) between cells on a regular basis, and that these TEs can transpose, multiply, and invade receiving cells. These conclusions are based on a series of experiments consisting in releasing cfChPs isolated from various human sera into the culture medium of mouse cells, and using FISH and immunofluorescence to monitor the state and fate of cfChPs after several passages of the mouse cell line.Strengths:The results presented in this study are interesting because they may reveal unsuspected properties of some cell types that may be able to internalize free-circulating chromatin, leading to its chromosomal incorporation, expression, and unleashing of TEs. The authors propose that this phenomenon may have profound impacts in terms of diseases and genome evolution. They even suggest that this could occur in germ cells, leading to within-organism HGT with long-term consequences.

Again the reviewer makes the same mistake. We do not claim that the internalized cfChPs are incorporated into the chromosomes. We have addressed this issue above.

We have a feeling that the reviewer has not understood our work – which is the discovery of “satellite genomes” which function autonomously outside the nuclear genome.

Weaknesses:The claims of massive HGT between cells through internalization of cfChPs are not well supported because they are only based on evidence from one type of methodological approach: immunofluorescence and fluorescent in situ hybridization (FISH) using protein antibodies and DNA probes. Yet, such strong claims require validation by at least one, but preferably multiple, additional orthogonal approaches. This includes, for example, whole genome sequencing (to validate concatemerization, integration in receiving cells, transposition in receiving cells), RNA-seq (to validate expression), ChiP-seq (to validate chromatin state).

We disagree with the reviewer that our study is incomplete because we did not perform whole genome sequencing. Tens of thousands of genomes have been sequenced, and yet they have failed to detect the presence of the numerous “satellite genomes” that we describe in our paper. To that extent whole genome sequencing has proved to be an inappropriate approach. Rather, the reviewer should have commended us for the numerous control experiments that we have done to ensure that our FISH probes and antibodies are target specific and do not cross-react.

Should HGT through internalization of circulating chromatin occur on a massive scale, as claimed in this study, and as illustrated by the many FISH foci observed on Fig 3 for example, one would expect that the level of somatic mosaicism may be so high that it would prevent assembling a contiguous genome for a given organism. Yet, telomere-to-telomere genomes have been produced for many eukaryote species, calling into question the conclusions of this study.

The reviewer has raised a related issue below and we have responded to both of them together.

**Recommendations for the authors:**

**Reviewer #1 (Recommendations for the authors):**
I thank the authors for taking my comments and those of the other reviewer into account and for adding new material to this new version of the manuscript. Among other modifications/additions, they now mention that they think that NIH3T3 cells treated with cfChPs die out after 250 passages because of genomic instability which might be caused by horizontal transfer of cfChPs DNA into the genome of treated cells (pp. 45-46, lines 725-731). However, no definitive formal proof of genomic instability and horizontal transfer is provided.

We mention that the NIH3T3 cells treated with cfChPs die out after 250 passages in response to the reviewer’s earlier comment “Should HGT through internalization of circulating chromatin occur on a massive scale, as claimed in this study, and as illustrated by the many FISH foci observed in Fig 3 for example, one would expect that the level of somatic mosaicism may be so high that it would prevent assembling a contiguous genome for a given organism”.

We have agreed with the reviewer and have simply speculated that the cells may die because of extreme genomic instability. We have left it as a speculation without diverting our paper in a different direction to prove genomic instability.

The authors now refer to an earlier study they conducted in which they Illumina-sequenced NIH3T3 cells treated with cfChPs (pp. 48, lines. 781-792). This study revealed the presence of human DNA in the mouse cell culture. However, it is unclear to me how the author can conclude that the human DNA was inside mouse cells (rather than persisting in the culture medium as cfChPs) and it is also unclear how this supports horizontal transfer of human DNA into the genome of mouse cells. Horizontal transfer implies integration of human DNA into mouse DNA, through the formation of phosphodiester bounds between human nucleotides and mouse nucleotides. The previous Illumina-sequencing study and the current study do not show that such integration has occured. I might be wrong but I tend to think that DNA FISH signals showing that human DNA lies next to mouse DNA does not necessarily imply that human DNA has integrated into mouse DNA. Perhaps such signals could result from interactions at the protein level between human cfChPs and mouse chromatin?

With due respect, our earlier genome sequencing study that the reviewer refers to was done on two single cell clones developed following treatment with cfChPs. So, the question of cfChPs lurking in the culture medium does not arise.

The authors should be commended for doing so many FISH experiments. But in my opinion, and as already mentioned in my earlier review of this work, horizontal transfer of human DNA into mouse DNA should first be demonstrated by strong DNA sequencing evidence (multiple long and short reads supporting human/mouse breakpoints; discarding technical DNA chimeras) and only then eventually confirmed by FISH.

As mentioned earlier, we disagree with the reviewer that our study is incomplete because we did not perform whole genome sequencing. Tens of thousands of genomes have been sequenced, and yet they have failed to detect the presence of the numerous “satellite genomes” that we describe in our paper. To that extent whole genome sequencing has proved to be an inappropriate approach. Rather, the reviewer should have commended us for the numerous control experiments that we have done to ensure that our FISH probes and antibodies are target specific and do not cross-react.

Regarding my comment on the quantity of human cfChPs that has been used for the experiments, the authors replied that they chose this quantity because it worked in a previous study. Could they perhaps explain why they chose this quantity in the earlier study? Is there any biological reason to choose 10 ng and not more or less? Is 10 ng realistic biologically? Could it be that 10 ng is orders of magnitude higher than the quantity of cfChPs normally circulating in multicellular organisms and that this could explain, at least in part, the results obtained in this study?

The reviewer again raises the same issue to which we have already addressed in our revised manuscript. To quote “We chose to use 10ng based on our earlier report in which we had obtained robust biological effects such as activation of DDR and activation of apoptotic pathways using this concentration of cfChPs (Mittra I et. al., 2015)”.

It is also mentioned in the response that RNA-seq has been performed on mouse cells treated with cfChPs, and that this confirms human-mouse fusion (genomic integration). Since these results are not included in the manuscript, I cannot judge how robust they are and whether they reflect a biological process rather than technical issues (technical chimeras formed during the RNA-seq protocol is a well-known artifact). In any case, I do not think that genomic integration can be demonstrated through RNA-seq as junction between human and mouse RNA could occur at the RNA level (i.e. after transcription). RNA-seq could however show whether human-mouse chimeras that have been validated by DNA-sequencing are expressed or not.

We did perform transcriptome sequencing as suggested earlier by the reviewer, but realized that the amount of material required to be incorporated into the manuscript to include “material and methods”, “results”, “discussion”, “figures” and “legends to figures” and “supplementary figures and tables” would be so massive that it will detract from the flow of our work and hijack it in a different direction. We have, therefore, decided to publish the transcriptome results as a separate manuscript.

Given these comments, I believe that most of the weaknesses I mentioned in my review of the first version of this work still hold true.An important modification is that the work has been repeated in other cell lines, hence I removed this criticism from my earlier review.

Additional changes made

We have now rewritten the “Abstract” to 250 words to fit in eLife’s instructions. It was not possible to reduce the word count further.We have provided the Video 1 as separate file instead of link.Some of Figure Supplements (which were stand-alone) are now given as main figures. We have re-arranged Figures and Figure Supplements in accordance with eLife’s instructions.We have now provided a list of the various cell lines used in this study, their tissue origin and procurement source in Supplementary File 3.

[The following is the authors’ response to the original reviews.]

**Public Reviews:**

**Reviewer #1 (Public review):**
Summary:Horizontal gene transfer is the transmission of genetic material between organisms through ways other than reproduction. Frequent in prokaryotes, this mode of genetic exchange is scarcer in eukaryotes, especially in multicellular eukaryotes. Furthermore, the mechanisms involved in eukaryotic HGT are unknown. This article by Banerjee et al. claims that HGT occurs massively between cells of multicellular organisms. According to this study, the cell free chromatin particles (cfChPs) that are massively released by dying cells are incorporated in the nucleus of neighboring cells. These cfChPs are frequently rearranged and amplified to form concatemers, they are made of open chromatin, expressed, and capable of producing proteins. Furthermore, the study also suggests that cfChPs transmit transposable elements (TEs) between cells on a regular basis, and that these TEs can transpose, multiply, and invade receiving cells. These conclusions are based on a series of experiments consisting in releasing cfChPs isolated from various human sera into the culture medium of mouse cells, and using FISH and immunofluorescence to monitor the state and fate of cfChPs after several passages of the mouse cell line.Strengths:The results presented in this study are interesting because they may reveal unsuspected properties of some cell types that may be able to internalize free-circulating chromatin, leading to its chromosomal incorporation, expression, and unleashing of TEs. The authors propose that this phenomenon may have profound impacts in terms of diseases and genome evolution. They even suggest that this could occur in germ cells, leading to within-organism HGT with long-term consequences.Weaknesses:The claims of massive HGT between cells through internalization of cfChPs are not well supported because they are only based on evidence from one type of methodological approach: immunofluorescence and fluorescent in situ hybridization (FISH) using protein antibodies and DNA probes. Yet, such strong claims require validation by at least one, but preferably multiple, additional orthogonal approaches. This includes, for example, whole genome sequencing (to validate concatemerization, integration in receiving cells, transposition in receiving cells), RNA-seq (to validate expression), ChiP-seq (to validate chromatin state).

We have responded to this criticism under “Reviewer #1 (Recommendations for the authors, item no. 1-4)”.

Another weakness of this study is that it is performed only in one receiving cell type (NIH3T3 mouse cells). Thus, rather than a general phenomenon occurring on a massive scale in every multicellular organism, it could merely reflect aberrant properties of a cell line that for some reason became permeable to exogenous cfChPs. This begs the question of the relevance of this study for living organisms.

We have responded to this criticism under “Reviewer #1 (Recommendations for the authors, item no. 6)”.

Should HGT through internalization of circulating chromatin occur on a massive scale, as claimed in this study, and as illustrated by the many FISH foci observed in Fig 3 for example, one would expect that the level of somatic mosaicism may be so high that it would prevent assembling a contiguous genome for a given organism. Yet, telomere-to-telomere genomes have been produced for many eukaryote species, calling into question the conclusions of this study.

The reviewer is right in expecting that the level of somatic mosaicism may be so high that it would prevent assembling a contiguous genome. This is indeed the case, and we find that beyond ~ 250 passages the cfChPs treated NIH3T3 cells begin to die out apparently become their genomes have become too unstable for survival. This point will be highlighted in the revised version (pp. 45-46, lines 725-731).

**Reviewer #2 (Public review):**
I must note that my comments pertain to the evolutionary interpretations rather than the study's technical results. The techniques appear to be appropriately applied and interpreted, but I do not feel sufficiently qualified to assess this aspect of the work in detail.I was repeatedly puzzled by the use of the term "function." Part of the issue may stem from slightly different interpretations of this word in different fields. In my understanding, "function" should denote not just what a structure does, but what it has been selected for. In this context, where it is unclear if cfChPs have been selected for in any way, the use of this term seems questionable.

We agree. We have removed the term “function” wherever we felt we had used it inappropriately.

Similarly, the term "predatory genome," used in the title and throughout the paper, appears ambiguous and unjustified. At this stage, I am unconvinced that cfChPs provide any evolutionary advantage to the genome. It is entirely possible that these structures have no function whatsoever and could simply be byproducts of other processes. The findings presented in this study do not rule out this neutral hypothesis. Alternatively, some particular components of the genome could be driving the process and may have been selected to do so. This brings us to the hypothesis that cfChPs could serve as vehicles for transposable elements. While speculative, this idea seems to be compatible with the study's findings and merits further exploration.

We agree with the reviewer’s viewpoint. We have replaced the term “predatory genome” with a more realistic term “satellite genome” in the title and throughout the manuscript. We have also thoroughly revised the discussion section and elaborated on the potential role of LINE-1 and Alu elements carried by the concatemers in mammalian evolution. (pp. 46-47, lines 743-756).

I also found some elements of the discussion unclear and speculative, particularly the final section on the evolution of mammals. If the intention is simply to highlight the evolutionary impact of horizontal transfer of transposable elements (e.g., as a source of new mutations), this should be explicitly stated. In any case, this part of the discussion requires further clarification and justification.

As mentioned above, we have revised the “discussion” section taking into account the issues raised by the reviewer and highlighted the potential role of cfChPs in evolution by acting as vehicles of transposable elements.

In summary, this study presents important new findings on the behavior of cfChPs when introduced into a foreign cellular context. However, it overextends its evolutionary interpretations, often in an unclear and speculative manner. The concept of the "predatory genome" should be better defined and justified or removed altogether. Conversely, the suggestion that cfChPs may function at the level of transposable elements (rather than the entire genome or organism) could be given more emphasis.

As mentioned above, we have replaced the term “predatory genome” with “satellite genome” and revised the “discussion” section taking into account the issues raised by the reviewer.

**Reviewer #1 (Recommendations for the authors):**
(1) I strongly recommend validating the findings of this study using other approaches. Whole genome sequencing using both short and long reads should be used to validate the presence of human DNA in the mouse cell line, as well as its integration into the mouse genome and concatemerization. Breakpoints between mouse and human DNA can be searched in individual reads. Finding these breakpoints in multiple reads from two or more sequencing technologies would strengthen their biological origin. Illumina and ONT sequencing are now routinely performed by many labs, such that this validation should be straightforward. In addition to validating the findings of the current study, it would allow performance of an in-depth characterization of the rearrangements undergone by both human cfChPs and the mouse genome after internalization of cfChPs, including identification of human TE copies integrated through bona fide transposition events into the mouse genome. New copies of LINE and Alu TEs should be flanked by target site duplications. LINE copies should be frequently 5' truncated, as observed in many studies of somatic transposition in human cells.(2) Furthermore, should the high level of cell-to-cell HGT detected in this study occur on a regular basis within multicellular organisms, validating it through a reanalysis of whole genome sequencing data available in public databases should be relatively easy. One would expect to find a high number of structural variants that for some reason have so far gone under the radar.(3) Short and long-read RNA-seq should be performed to validate the expression of human cfChPs in mouse cells. I would also recommend performing ChIP-seq on routinely targeted histone marks to validate the chromatin state of human cfChPs in mouse cells.(4) The claim that fused human proteins are produced in mouse cells after exposing them to human cfChPs should be validated using mass spectrometry.

The reviewer has suggested a plethora of techniques to validate our findings. Clearly, it is neither possible to undertake all of them nor to incorporate them into the manuscript.However, as suggested by the reviewer, we did conduct transcriptome sequencing of cfChPs treated NIH3T3 cells and were able to detect the presence of human-human fusion sequences (representing concatemerisation) as well as human-mouse fusion sequences (representing genomic integration). However, we realized that the amount of material required to be incorporated into the manuscript to include “material and methods”, “results”, “discussion”, “figures” and “legends to figures” and “supplementary figures and tables” would be so massive that it will detract from the flow of our work and hijack it in a different direction. We have, therefore, decided to publish the transcriptome results as a separate manuscript.However, to address the reviewer’s concerns we have now referred to results of our earlier whole genome sequencing study of NIH3T3 cells similarly treated with cfChPs wherein we had conclusively detected the presence of human DNA and human Alu sequences in the treated mouse cells. These findings have now been added as an independent paragraph (pp. 48, lines. 781-792).

(5) It is unclear from what is shown in the paper (increase in FISH signal intensity using Alu and L1 probes) if the increase in TE copy number is due to bona fide transposition or to amplification of cfChPs as a whole, through mechanisms other than transposition. It is also unclear whether human TEs end up being integrated into the neighboring mouse genome. This should be validated by whole genome sequencing.

Our results suggest that TEs amplify and increase their copy number due to their association with DNA polymerase and their ability to synthesize DNA (Figure 14a and b). Our study design cannot demonstrate transposition which will require real time imaging.

The possibility of incorporation of TEs into the mouse genome is supported by our earlier genome sequencing work, referred to above, wherein we detected multiple human Alu sequences in the mouse genome (pp. 48, lines. 781-792).

(6) In order to be able to generalize the findings of this study, I strongly encourage the authors to repeat their experiments using other cell types.

We thank the reviewer for this suggestion. We have now used four different cell lines derived from four different species and demonstrated that horizontal transfer of cfChPs occur in all of them suggesting that it is a universal phenomenon. (pp. 37, lines 560-572) and (Supplementary Fig. S14a-d).

We have also mentioned this in the abstract (pp. 3, lines 52-54).

(7) Since the results obtained when using cfChPs isolated from healthy individuals are identical to those shown when using cfChPs from cancer sera, I wonder why the authors chose to focus mainly on results from cancer-derived cfChPs and not on those from healthy sera.

Most of the experiments were conducted using cfChPs isolated from cancer patients because of our especial interest in cancer, and our earlier results (Mittra et al., 2015) which had shown that cfChPs isolated from cancer patients had significantly greater activity in terms of DNA damage and activation of apoptotic pathways than those isolated from healthy individuals. We have now incorporated the above justification on (pp. 6, lines. 124-128).

(8) Line 125: how was the 10-ng quantity (of human cfChPs added to the mouse cell culture) chosen and how does it compare to the quantity of cfChPs normally circulating in multicellular organisms?

We chose to use 10ng based on our earlier report in which we had obtained robust biological effects such as activation of DDR and apoptotic pathways using this concentration of cfChPs (Mittra I et. al. 2015). We have now incorporated the justification of using this dose in our manuscript (pp. 51-52, lines. 867-870).

(9) Could the authors explain why they repeated several of their experiments in metaphase spreads, in addition to interphase?

We conducted experiments on metaphase spreads in addition to those on chromatin fibres because of the current heightened interest in extra-chromosomal DNA in cancer, which have largely been based on metaphase spreads. We were interested to see how the cfChP concatemers might relate to the characteristics of cancer extrachromosomal DNA and whether the latter in fact represent cfChPs concatemers acquired from surrounding dying cancer cells. We have now mentioned this on pp. 7, lines 150-155.

(10) Regarding negative controls consisting in checking whether human probes cross-react with mouse DNA or proteins, I suggest that the stringency of washes (temperature, reagents) should be clearly stated in the manuscript, such that the reader can easily see that it was identical for controls and positive experiments.

We were fully aware of these issues and were careful to ensure that washing steps were conducted meticulously. The careful washing steps have been repeatedly emphasized under the section on “Immunofluorescence and FISH” (pp. 54-55, lines. 922-944).

(11) I am not an expert in Immuno-FISH and FISH with ribosomal probes but it can be expected that ribosomal RNA and RNA polymerase are quite conserved (and thus highly similar) between humans and mice. A more detailed explanation of how these probes were designed to avoid cross-reactivity would be welcome.

We were aware of this issue and conducted negative control experiment to ensure that the human ribosomal RNA probe and RNA polymerase antibody did not cross-react with mouse. Please see Supplementary Fig. S4c.

(12) Finally, I could not understand why the cfChPs internalized by neighboring cells are called predatory genomes. I could not find any justification for this term in the manuscript.

We agree and this criticism has also been made by #Reviewer 2. We have now replaced the term “predatory” genomes with “satellite” genomes.

**Reviewer #2 (Recommendations for the authors):**
(1) P2 L34: The term "role" seems to imply "what something is supposed to do" (similar to "function"). Perhaps "impact" would be more neutral. Additionally, "poorly defined" is vague-do you mean "unknown"?

We thank the reviewer for this suggestion. We have now rephrased the sentence to read “Horizontal gene transfer (HGT) plays an important evolutionary role in prokaryotes, but it is thought to be less frequent in mammals.” (pp. 2, lines. 26-27).

(2) P2 L35: It seems that the dash should come after "human blood."

Thank you, we have changed the position of the dash (pp. 2, line. 29).

(3) P2 L37: Must we assume these structures have a function? Could they not simply be side effects of other processes?

We think this is a matter of semantics, especially since we show that cfChPs once inside the cell perform many functions such as replication, DNA synthesis, RNA synthesis, protein synthesis etc. We, therefore, think the word “function” is not inappropriate.

(4) Abstract: After reading the abstract, I am unclear on the concept of a "predatory genome." Based on the summarized results, it seems one cannot conclude that these elements provide any adaptive value to the genome.

We agree. We have now replaced the term “predatory” genomes with a more realistic term viz. “satellite” genomes.

(5) Video abstract: The video abstract does not currently stand on its own and needs more context to be self-explanatory.

Thank you for pointing this out. We have now created a new and much more professional video with more context which we hope will meet with the reviewer’s approval.

(6) P4 L67: Again, I am uncertain that HGT should be said to have "a role" in mammals, although it clearly has implications and consequences. Perhaps "role" here is intended to mean "consequence"?

We have now changed the sentence to read as follows “However, defining the occurrence of HGT in mammals has been a challenge” (pp. 4, line. 73).

(7) P6 L111: The phrase "to obtain a new perspective about the process of evolution" is unclear. What exactly is meant by this statement?

We have replaced this sentence altogether which now reads “The results of these experiments are presented in this article which may help to throw new light on mammalian evolution, ageing and cancer” (pp. 5-6, lines 116-118).

(8) P38 L588: The term "predatory genome" has not been defined, making it difficult to assess its relevance.

This issue has been addressed above.

(9) P39 L604: The statement "transposable elements are not inherent to the cell" suggests that some TEs could originate externally, but this does not rule out that others are intrinsic. In other words, TEs are still inherent to the cell.

This part of the discussion section has been rewritten and the above sentence has been deleted.

(10) P39 L609: The phrase "may have evolutionary functions by acting as transposable elements" is unclear. Perhaps it is meant that these structures may serve as vehicles for TEs?

This sentence has disappeared altogether in the revised discussion section.

(11) P41 L643: "Thus, we hypothesize ... extensively modified to act as foreign genetic elements." This sentence is unclear. Are the authors referring to evolutionary changes in mammals in general (which overlooks the role of standard mutational processes)? Or is it being proposed that structural mutations (including TE integrations) could be mediated by cfChPs in addition to other mutational mechanisms?

We have replaced this sentence which now reads “Thus, “within-self” HGT may occur in mammals on a massive scale via the medium of cfChP concatemers that have undergone extensive and complex modifications resulting in their behaviour as “foreign” genetic elements” (pp. 47, lines 763-766).

(12) P41 L150: The paragraph beginning with "It has been proposed that extreme environmental..." transitions too abruptly from HGT to adaptation. Is it being proposed that cfChPs are evolutionary processes selected for their adaptive potential? This idea is far too speculative at this stage and requires clarification.

We agree. This paragraph has been removed.

(13) P43 L681: This summary appears overly speculative and unclear, particularly as the concept of a "predatory genome" remains undefined and thus cannot be justified. It suggests that cfChPs represent an alternative lifestyle for the entire genome, although alternative explanations seem far more plausible at this point.

We have now replaced the term “predatory” genome with “satellite” genome. The relevant part of the summary section has also been partially revised (pp. 49-50, lines 817-831).

Changes independent of reviewers’ comments.

We have made the following additions / modifications.

(1) The abstract has been modified and it’s “conclusion” section has been rewritten.

(2) Section 1.14 has been newly added together with accompanying Figures 15 a,b and c.

(3) The “Discussion” section has been greatly modified and parts of it has been rewritten.